# Multifactorial seroprofiling dissects the contribution of pre-existing human coronaviruses responses to SARS-CoV-2 immunity

Irene A. Abela [1,2,9], Chloé Pasin[1,2,9], Magdalena Schwarzmüller [1,9], Selina Epp[1], Michèle E. Sickmann[1], Merle M. Schanz[1], Peter Rusert[1], Jacqueline Weber[1], Stefan Schmutz [1], Annette Audigé[1], Liridona Maliqi[1], Annika Hunziker[1], Maria C. Hesselman[1], Cyrille R. Niklaus[1], Jochen Gottschalk[3], Eméry Schindler[3], Alexander Wepf[4], Urs Karrer[5], Aline Wolfensberger[2], Silvana K. Rampini [6], Patrick M. Meyer Sauteur [7], Christoph Berger[7], Michael Huber [1], Jürg Böni [1], Dominique L. Braun[1,2], Maddalena Marconato[8], Markus G. Manz [8], Beat M. Frey [3], Huldrych F. Günthard [1,2,10✉], Roger D. Kouyos [1,2,10✉] & Alexandra Trkola [1,10✉]

Determination of SARS-CoV-2 antibody responses in the context of pre-existing immunity to circulating human coronavirus (HCoV) is critical for understanding protective immunity. Here we perform a multifactorial analysis of SARS-CoV-2 and HCoV antibody responses in pre-pandemic ($N = 825$) and SARS-CoV-2-infected donors ($N = 389$) using a custom-designed multiplex ABCORA assay. ABCORA seroprofiling, when combined with computational modeling, enables accurate definition of SARS-CoV-2 seroconversion and prediction of neutralization activity, and reveals intriguing interrelations with HCoV immunity. Specifically, higher HCoV antibody levels in SARS-CoV-2-negative donors suggest that pre-existing HCoV immunity may provide protection against SARS-CoV-2 acquisition. In those infected, higher HCoV activity is associated with elevated SARS-CoV-2 responses, indicating cross-stimulation. Most importantly, HCoV immunity may impact disease severity, as patients with high HCoV reactivity are less likely to require hospitalization. Collectively, our results suggest that HCoV immunity may promote rapid development of SARS-CoV-2-specific immunity, thereby underscoring the importance of exploring cross-protective responses for comprehensive coronavirus prevention.

[1] Institute of Medical Virology, University of Zurich, Zurich, Switzerland. [2] Division of Infectious Diseases and Hospital Epidemiology, University Hospital Zurich, Zurich, Switzerland. [3] Blood Transfusion Service Zurich, Zurich, Switzerland. [4] Institute of Laboratory Medicine, Cantonal Hospital Winterthur, Winterthur, Switzerland. [5] Department of Medicine, Cantonal Hospital Winterthur, Winterthur, Switzerland. [6] Department of Internal Medicine, University Hospital Zurich, Zurich, Switzerland. [7] Division of Infectious Diseases and Hospital Epidemiology, University Children's Hospital Zurich, Zurich, Switzerland. [8] Department of Medical Oncology and Hematology, University Hospital and University of Zurich, Zurich, Switzerland. [9] These authors contributed equally: Irene A. Abela, Chloé Pasin, Magdalena Schwarzmüller. [10] These authors jointly supervised: Huldrych F. Günthard, Roger D. Kouyos, Alexandra Trkola. ✉email: Huldrych.Guenthard@usz.ch; Roger.Kouyos@usz.ch; trkola.alexandra@virology.uzh.ch

Monitoring the antibody response to SARS-CoV-2 is critical to define correlates of vaccine protection, differences in susceptibility to infection and in disease severity. The picture of the antibody landscape to SARS-CoV-2 that has thus far evolved is complex. The antibody response to SARS-CoV-2 is rapid, and triggers strong IgM, IgA and IgG responses[1,2]. Both binding and neutralizing responses increase with disease severity and show in part dependence on demographic parameters such as age and gender[3–5]. It remains, however, unclear which factors are independent drivers of antibody responses, reflect severe disease courses or are confounded by other factors including infection length and comorbidities. Waning IgG binding and neutralizing antibody titers may be particularly pronounced in individuals with asymptomatic or mild infection[6–9]. IgG responses to spike (S) glycoprotein may persist longer than to nucleocapsid protein (N)[7,10,11] and can in part undergo affinity maturation post virus clearance[5]. Current serological analyses predominantly focus on measuring reactivity to N, the spike glycoprotein S1 subunit and the ACE2 receptor-binding domain (RBD) in S1[2,5,12–16]. Antibodies to RBD and the receptor-binding motif within the RBD constitute the main group of neutralizing antibodies, followed by S1 trimer specific, spike N-terminal domain, and spike S2 neutralizing antibodies[16–22]. S1 and RBD binding correlate with neutralizing activity in both natural and vaccine-induced immune responses providing means to estimate the potential for neutralization where neutralization capacity cannot be assessed directly[6,8,10]. Considering the complex antibody response patterns, possibilities to capture the dynamics of the SARS-CoV-2 response across diverse Immunoglobulin (Ig) classes and SARS-CoV-2 antigens are needed to ascertain sensitive detection of seroconversion and sero-reversion and to establish links to protective, neutralizing activity post infection and post vaccination.

Infections with circulating human coronaviruses (HCoV), alphacoronavirus (HCoV-229E, HCoV-NL63) and betacoronavirus (HCoV-HKU1, HCoV-OC43), are common and contribute considerably to the seasonal respiratory disease burden in humans[23,24]. Despite an overall modest sequence homology between SARS-CoV-2 and circulating HCoVs, several conserved regions exist and antibody cross-reactivity may occur[25–27]. While dismissed in the diagnostic setting as false-positives[28], cross-reactive antibodies may bear biological relevance as suggested for SARS-CoV-2 S2 cross-neutralizing antibodies[29]. Uncertainty remains, however, whether cross-reactive HCoV antibody responses influence the evolution of SARS-CoV-2 specific immunity. Positive impact by providing early low affinity memory responses to build on and mature as well as negative influences following the antigenic sin principle[30] by boosting nonprotective cross-reactive antibodies on the expense of de novo responses can be envisaged. Of particular note, cross-reactive HCoV T helper cell responses were shown to positively impact SARS-CoV-2 specific immunity[31]. In view of this, the definition of pre-existing immunity due to prior infection with HCoVs will become important in clinical diagnosis and strategies to record and unveil the complex interdependencies HCoV and SARS-CoV-2 responses side by side are needed to fill this knowledge gap.

Here we report on the development of a serological assay that allows multifactorial seroprofiling of SARS-CoV-2 and HCoV responses at high diagnostic accuracy. Seroprofiling of a large cohort of SARS-CoV-2 infected and uninfected individuals provided key insights into the interdependencies of HCoV and SARS-CoV-2 antibody responses. The results highlight a potential protective role of HCoV-specific responses in SARS-CoV-2 acquisition as well as in shaping the SARS-CoV-2 response upon infection.

## Results

**Multifactorial seroprofiling defines SARS-CoV-2 specific responses.** Recognizing the need for comprehensive SARS-CoV-2 serological profiling to elucidate central questions in SARS-CoV-2 immunity and its interdependencies with HCoV responses, we created a bead-based multiplex immunoassay to measure specific IgG, IgA and IgM responses to SARS-CoV-2 RBD, S1, S2 and N (Supplementary Fig. 1). The assay records in total 12 SARS-CoV-2 specific antibody parameters (4 antigens across 3 Ig classes) with high diagnostic accuracy (see methods, Supplementary Figs. 1–3 and Supplementary Tables 1, 2) and further includes the S1 protein of HCoV-HKU1 to screen cross-reactive antibodies alongside SARS-CoV-2 responses. According to the test's design to monitor antibodies to two coronaviruses, we termed the assay AntiBody CORonavirus Assay (ABCORA) 2.0.

Measurements in ABCORA are expressed as median fluorescence intensity (MFI) corrected for background binding (fold over empty beads, FOE). To distinguish SARS-CoV-2-specific from cross-reactive antibodies, we defined MFI-FOE thresholds for each of the 12 SARS-CoV-2 antigen and Ig class combinations based on plasma antibody reactivity in training cohorts of pre-pandemic healthy donors (Training I, $N = 573$), donors with recent HCoV infection (Training II, $N = 75$) and donors with confirmed SARS-CoV-2 infection (Training III, $N = 175$) (Fig. 1a, Supplementary Table 3). Positive call criteria were defined to ascertain that in at least two of the 12 antigen and Ig combinations the threshold is reached (Supplementary Table 4). The final threshold and positive call criteria allowed for a differentiation of partial (only IgM and IgA responses) to full seroconversion (including IgG responses). In addition, the criteria denote samples with weak reactivity and/or indeterminate reactivity (Supplementary Table 5).

Pre-pandemic patients with documented, recent HCoV infection (Training II, $N = 75$; OC43 ($N = 27$), HKU1 ($N = 17$), NL63 ($N = 22$), 229E ($N = 9$)) comprised individuals with different underlying severe diseases including immune compromised patients that underwent diagnostic screening for HCoV. HCoV specific activity was overall lower in this syndromic group but showed, as expected, enriched HCoV reactivity against the infecting HCoV (Supplementary Fig. 4). Importantly, we observed no indication of cross-reactivity with SARS-CoV-2 antigens that affects the ABCORA readout (Fig. 1a, b). Considering data of all training cohorts (I–III), ABCORA 2.0 exhibited a high sensitivity and specificity, reaching 94.29% sensitivity and 99.07% specificity (Fig. 1c, Supplementary Table 3).

To enable an analysis of cross-reactivities and interdependencies between SARS-CoV-2 and HCoV antibodies, we recorded reactivity to the S1 unit of HCoV-HKU1 in addition to the SARS-CoV-2 antigens (Fig. 1a). Owing to the high prevalence of HCoV antibodies and the ensuing lack of true-negative controls, we set no thresholds to rate HKU1 reactivity as positive/negative. Overall, SARS-CoV-2 cross-reactivity was low in pre-pandemic samples despite notable HKU1 activity (Fig. 1a). Correlation analysis revealed modest interdependencies of SARS-CoV-2 and HKU1 plasma antibody reactivity in SARS-CoV-2 positive and pre-pandemic donors. This predominantly involved IgM responses, with individuals with recent HCoV infection showing the highest correlation in IgM for HKU1 and SARS-CoV-2 activity (Supplementary Fig. 5). These data underline that a low level of cross-reactive activity exists that needs to be respected in assay design, analysis and validation.

Verification of ABCORA 2.0 on separate validation cohorts of pre-pandemic healthy adults ($N = 252$), pre-pandemic children ($N = 169$) and individuals with documented SARS-CoV-2 infection ($N = 214$) (Fig. 1a, b, Supplementary Table 3) confirmed the validity of the chosen assay criteria. Combining training and

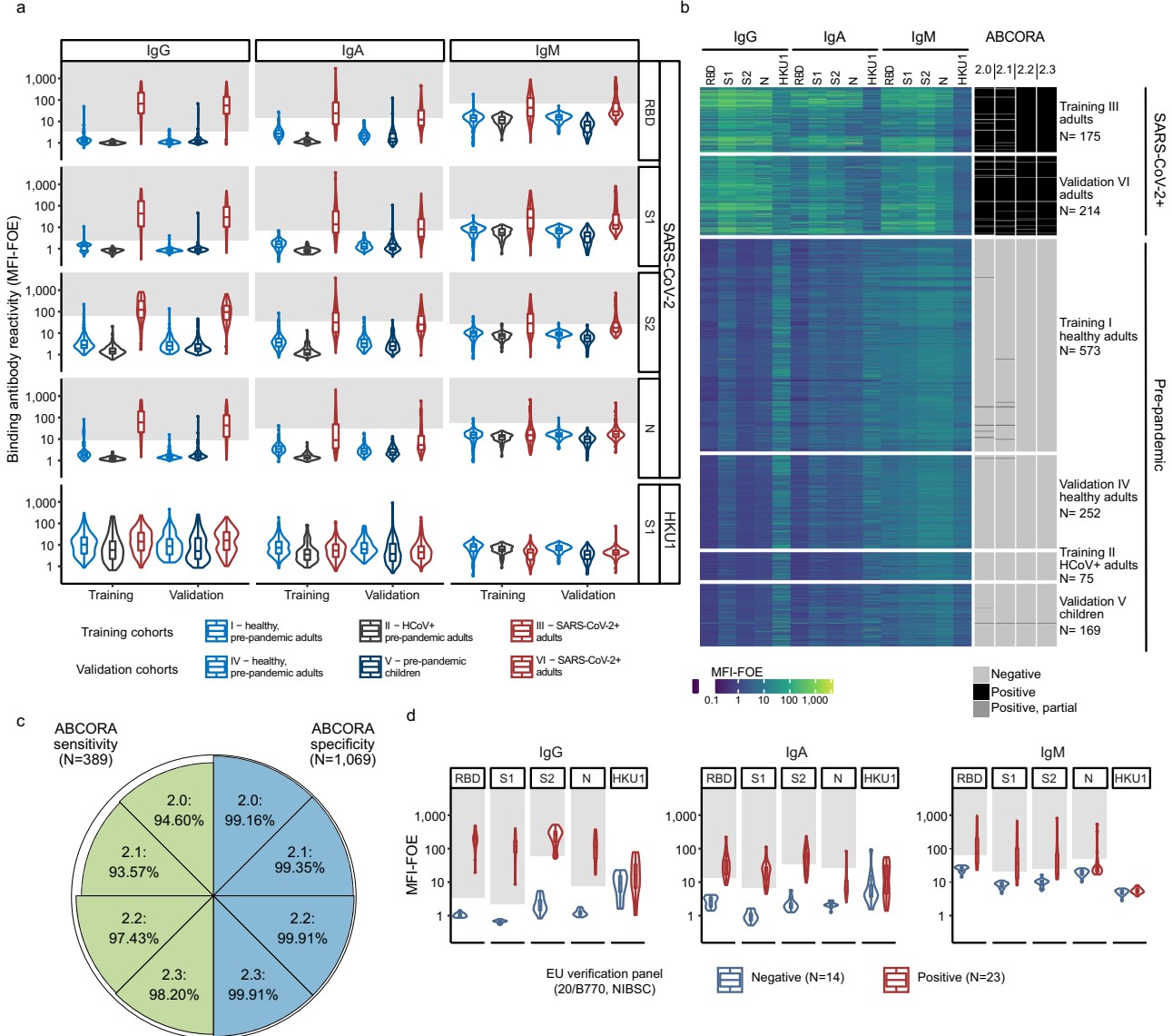

**Fig. 1 Seroprofiling SARS-CoV-2 responses. a** Assessment of the multiplex SARS-CoV-2 ABCORA 2.0 on the indicated training ($N = 823$) and validation ($N = 635$) cohorts (Supplementary Table 3). Depicted are MFI signals normalized to empty bead controls (MFI-FOE). Gray boxes indicate values above the individually set MFI-FOE cut-offs for SARS-CoV-2 specific responses for each antigen (see Supplementary Table 4). Boxplots represent the following: median with the middle line, upper and lower quartiles with the box limits, 1.5x interquartile ranges with the whiskers. **b** Heatmap representing the measured MFI-FOE values and the outcomes predicted with ABCORA 2.0–2.3 of training and validation cohort measurements shown in (**a**). Negative, Positive, and Positive, partial refer to ranking according to ABCORA 2.0 as specified in Supplementary Table 5. **c** Sensitivity and specificity of ABCORA 2 assay versions based on the combined training and validation cohort data depicted in (**a**). Proportion of false negative samples (sensitivity; green) and proportion of false positive samples (specificity; blue) are represented by the reduction from 100% (outer circle) per segment. **d** Assessment of ABCORA 2.0 with the National Institute for Biological Standards and Control (NIBSC) Anti SARS-CoV-2 Verification Panel (20/B770) comprising SARS-CoV-2 positive (red) and negative (blue) panel serum samples. Gray boxes indicate values above the ABCORA 2.0 MFI-FOE cut-offs for SARS-CoV-2 specific responses for individual antigen-Ig combinations. Boxplots represent the following: median with the middle line, upper and lower quartiles with the box limits, 1.5x interquartile ranges with the whiskers.

validation cohorts of SARS-CoV-2 positive individuals ($N = 389$) and negative controls ($N = 825$), ABCORA 2.0 achieved a sensitivity of 94.60% and a specificity of 99.16% (Fig. 1c).

Of note, when analyzing children and adults in the validation cohort separately, we observed a slightly lower specificity amongst children (98.82%) compared to adults (99.60%), raising the possibility that cross-reactive activity in children may be more prevalent than in adults. Indeed, pre-pandemic children showed a higher correlation of IgM HKU1 and SARS-CoV-2 (Supplementary Fig. 5c), highlighting that interpretation of IgM SARS-CoV-2 activity can be complex.

**Computational analyses maximize specificity and sensitivity of SARS-CoV-2 seroprofiling.** To further increase specificity of the readout, we next explored two computational analysis extensions, a logistic regression model (ABCORA 2.1) and a random forest model (ABCORA 2.2). Both analysis strategies were established on the identical training dataset (Training I–III) used for the setup of ABCORA 2.0. Instead of obtaining 12 individual thresholds (one per antigen and Ig class), the computational models solely estimate the probability of a sample to be positive by providing a composite result across all 12 measurements and ranking sera positive or negative (1, 0 classification). For the

logistic regression ABCORA 2.1, we grouped SARS-CoV-2 binding activities displaying high correlation (Supplementary Fig. 6a) and included the mean value of their MFI-FOEs in the model. The random forest model ABCORA 2.2 included all 12 SARS-CoV-2 responses measured and aggregated the result of 1000 classification trees. On the combined training and validation cohorts of SARS-CoV-2 positive individuals ($N = 389$) and negative controls ($N = 825$), ABCORA 2.2 achieved a striking sensitivity of 97.43% and a specificity of 99.91% outperforming both ABCORA 2.0 and 2.1. Of note, positive calling by ABCORA 2.2 was dominated by IgG responses (Supplementary Fig. 6b).

We next explored whether incorporation of HKU1 reactivity into the random forest model may further improve the calling specificity and sensitivity. Indeed, a model that included HKU1 S1 as additional variable (ABCORA 2.3) increased sensitivity from 97.43% in ABCORA 2.2 to 98.20% (Fig. 1c, Supplementary Table 3) without reduction of the specificity. A sensitivity cross-validation analysis with randomized training and validation set confirmed the performance of ABCORA 2.3 (Supplementary Table 6). Owing to its combined high sensitivity and specificity, we therefore selected ABCORA 2.3 as the analysis strategy for rating global SARS-CoV-2 seroconversion.

We next verified the accuracy of ABCORA 2.0 and 2.3 in defining positive and negative SARS-CoV-2 immune status utilizing the National Institute for Biological Standards and Control (NIBSC) Anti SARS-CoV-2 Verification Panel (20/B770)[32]. This verification panel for serology assays includes 23 positive and 14 negative serum samples and allows direct comparison with other test systems[32]. Both ABCORA versions showed 100% sensitivity and 100% specificity on the verification plasma panel and compared favorably to commercial assay systems (Fig. 1d, Supplementary Table 7). To cross-reference these external verification results, we next compared the sensitivity of the ABCORA tests and three commercial serology test systems on a subset of the SARS-CoV-2 positive training cohort (cohort III, $N = 171$). Assays targeting the N protein (Elecsys® Anti-SARS-CoV-2 (Roche Diagnostics GmbH)), the RBD region of the S protein (Elecsys® Anti-SARS-CoV-2 S assay (Roche Diagnostics GmbH)), and the S1 subunit (EUROIMMUN Anti-SARS-CoV-2 ELISA (IgG)) were included. The results confirmed the analysis on the international NIBSC 20/B770 plasma panel, with ABCORA 2.0 and ABCORA 2.3 showing the highest sensitivity amongst the tested assays (Supplementary Table 8).

We thus conclude that ABCORA 2.0 seroprofiling in combination with ABCORA 2.3 defines positivity with the highest specificity and sensitivity. The individual antigen response evaluation by ABCORA 2.0 defines the stage of seroconversion status based on individual IgM, IgA and IgG cut-off values and thereby complements and maximizes the information that can be obtained by ABCORA 2 seroprofiling.

**Predicting SARS-CoV-2 neutralization based on ABCORA seroprofiling.** Determining neutralization activity is critical to gauge protective immunity. While neutralization can be directly measured with a range of authentic virus or pseudovirus SARS-CoV-2 neutralization tests[5,22,33], applying direct binding or competition tests as surrogate for neutralization activity remains of high interest for diagnostic purposes where cell-based assays are more difficult to implement[33,34]. In particular, S1 and RBD binding and ACE2 competition have been shown to correlate well with neutralization activity[5,8,10,19,33–37]. To explore neutralization predictors based on ABCORA 2.0, we probed in a first step the capacity of ABCORA to derive quantitative S1 and RBD readouts in a subset of SARS-CoV-2 positive patients ($N = 72$). ABCORA

2.0 measurements of serially diluted plasma were conducted to derive 50% effective concentrations (EC50, expressed as reciprocal plasma dilution) and area under the curve values (AUC expressed as MFI) for all 12 SARS-CoV-2 parameters (Fig. 2a, b). In addition, we quantified SARS-CoV-2 RBD and S1 responses via the RBD specific mAb CR3022[38] (Fig. 2c) and the WHO International Standard Anti-SARS-CoV-2 Immunoglobulin NIBSC 20/136[32] (Supplementary Fig. 7b, Supplementary Data 1). For this we quantified the respective antibody content of a positive control SARS-CoV-2 donor pool included in all ABCORA measurements and expressed the antibody content of individual plasma samples in relation to it (Fig. 2c, Supplementary Fig. 7). We then probed which of the ABCORA quantitative readouts correlated best with each other, the basic readout of ABCORA 2.0 (MFI-FOE at plasma dilution 1/100), and the quantitative Roche Elecsys S test (U/ml) (Fig. 2c, Supplementary Fig. 7). In addition to the individual antigen parameters, we also considered cumulative response values. These were total spike reactivity (sum of RBD, S1, S2 across all Ig classes), Ig class spike reactivity (sum of S1, RBD, S2 for one isotype) and antigen specific reactivity (sum of all Ig classes for one antigen). We observed a genuinely good correlation across the diverse spike parameters tested (Fig. 2d). The notable exception were classical EC50 values, which showed no to weak correlation across all parameters including the commercial test. Interestingly, AUC values, which in contrast to EC50 are a composite measure of concentration and signal strength, performed well. Of note, we observed highly variable SARS-CoV-2 antibody dose response curves, reaching in our cohort individual plateaus over a 4-log range (Supplementary Fig. 7). These plateaus are respected in the AUC readout, and are also recorded by the basic MFI-FOE ABCORA readout at 1/100 plasma dilution, but are not considered in EC50 determinations. Indeed the basic MFI-FOE showed a high correlation with the quantitative readouts across the tested variables including the quantitative commercial Roche Elecsys S test.

Based on these results, we concluded that the MFI-FOE readout solely at the 1/100 plasma dilution provides a highly reliable estimate for the S1 and RBD antibody content in plasma that can be used as a proxy for quantification without the need to titrate samples. We therefore employed the basic MFI-FOE in a next step to define neutralization predictors.

Neutralization activity to Wuhan-Hu-1 in SARS-CoV-2 positive individuals ($N = 467$) using an established SARS-CoV-2 pseudovirus neutralization test[5,22,33] revealed a broad range of 50% neutralizing titers (NT50) (post positive RT-PCR Fig. 3a ($N = 369$), post onset of symptoms, Supplementary Fig. 8a ($N = 333$)), in line with previous findings[5,8]. Early in infection (within 30 days of positive RT-PCR) neutralization titers were significantly higher ($p < 0.001$) and correlated better with binding parameters. As expected, IgG responses to spike antigens showed the highest correlation with neutralization activity (Fig. 3b, Supplementary Figs. 8, 9). We next grouped patients based on the population into high (NT50 > 250, $N = 332$) and no or low neutralizers (NT50 < 250, $N = 135$)[39] (Fig. 4a) and compared the prediction ability of six different classification models to assign individuals based on their ABCORA 2.0 binding patterns to these groups. Univariable logistic regression (ULR) models included only one variable: either the mean of MFI-FOE S1 reactivities (ULR-S1), or the mean of MFI-FOE RBD reactivities (ULR-RBD). A multivariable logistic regression (MLR) included both S1 and RBD mean reactivities. The additional models included all 12 antigen reactivities measured in ABCORA and comprised a random forest approach and two MLR strategies based on principle component analysis (PCA, 2 and 4 first axis). Models were compared based on AUC and the BIC (Bayesian

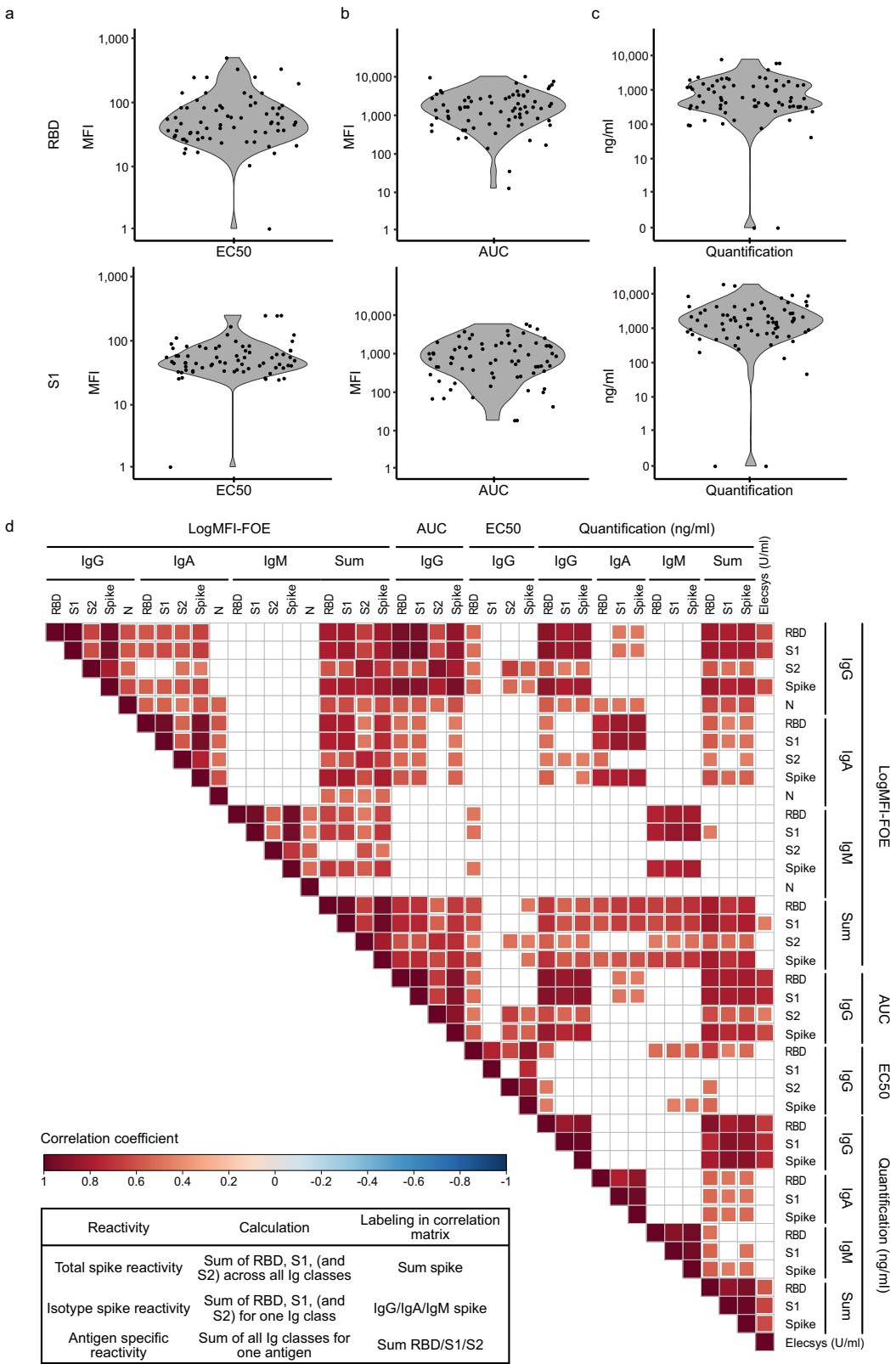

| Reactivity | Calculation | Labeling in correlation matrix |
|---|---|---|
| Total spike reactivity | Sum of RBD, S1, (and S2) across all Ig classes | Sum spike |
| Isotype spike reactivity | Sum of RBD, S1, (and S2) for one Ig class | IgG/IgA/IgM spike |
| Antigen specific reactivity | Sum of all Ig classes for one antigen | Sum RBD/S1/S2 |

information criterion[40]) by cross validation (Fig. 4b, c). All models performed similarly, with the univariable model based on the mean of S1 reactivities (ULR-S1) yielding the best BIC value. Receiver operating curve (ROC) analysis based on ULR-S1 showed a good capacity in predicting neutralization status yielding AUC 0.90 ($N = 467$, Fig. 4d). Exploring different cutoffs to balance sensitivity and specificity keeping both above 80%, we

chose to assign samples to the high neutralizers group if its predicted probability was above 70%. This corresponds to an 83% specificity in correctly assigning non–neutralizers and 80% sensitivity in assigning neutralizers (Fig. 4d, e). To increase the utility of the ULR-S1 prediction model for clinical diagnostics, we devised a modified neutralization prediction model ULR-S1-SOC based on the SOC values reported for ABCORA 2.0. At 70%

**Fig. 2 Quantification of SARS-CoV-2 specific antibody responses. a–c** Distribution of (**a**) 50% effective concentrations (EC50; expressed as reciprocal plasma dilution) and (**b**) area under the curve values (AUC; expressed as MFI) of titrated plasma from SARS-CoV-2 positive adults ($N = 72$) measured with ABCORA 2.0. **c** Titrated SARS-CoV-2 RBD and S1 responses were quantified using the RBD specific monoclonal antibody CR3022 (produced as IgG, IgA and IgM; expressed as ng/ml) as external standard. See Supplementary Fig. 7 for additional quantification with the WHO International Standard Anti-SARS-CoV-2 Immunoglobulin. **d** Spearman correlation matrix assessing agreement between ABCORA 2.0 based quantification readouts (EC50, AUC, RBD Ab standardized), the basic MFI-FOE measured at 1/100 plasma dilution (log), indicated summed logMFI-FOE values (1/100 dilution), and Roche Elecsys Anti-SARS-CoV-2 (S) assay results (U/ml). Nonsignificant correlations are left blank. Levels of significance are assessed by a two-sided test on the asymptotic t approximation of Spearman's rank correlation, and corrected by the Bonferroni method for multiple testing ($p < 0.05/780$). Color shading denotes correlation coefficient.

predicted probability, ULR-S1-SOC delivers neutralization prediction at similar sensitivity (81%) and specificity (81%) by examining if the composite S1 SOC value (sum of S1 SOC values for IgG, IgA and IgM) is below or above 9.7 (Fig. 4f). Correspondingly, a S1 SOC value above 17.3 corresponds to sensitivity 67% and specificity 94%. Of note, the interrelations between neutralization and S1 levels were equally apparent when we probed a lower cut-off of neutralization (NT50 < 100) (Supplementary Fig. 10, Supplementary Table 9). We therefore conclude that the basic SOC readout in ABCORA 2.0 can deliver a reliable prediction of high neutralization activity.

**Resolution of temporal antibody dynamics by ABCORA seroprofiling.** Cross-sectional analysis of antibody reactivity post SARS-CoV-2 diagnosis by RT-PCR ($N = 369$) and post onset of symptoms ($N = 333$) underlined the capacity of ABCORA seroprofiling to dissect onset, peak and waning of SARS-CoV-2 antibody responses (Fig. 5a, b, Supplementary Fig. 11). In individuals with known date of first SARS-CoV-2 RT-PCR diagnosis or onset of symptoms, ABCORA 2.3 detected early seroconversion in 98% (48 of 49) and 100% (9 of 9) of individuals within 7 days post RT-PCR and onset of symptoms, respectively. Besides IgM and IgA reactivity, IgG responses were readily detectable in ABCORA 2.0 after a few days of infection (Supplementary Fig. 11a).

Longitudinal assessment of a cohort of convalescent patients up to 11 months post infection (251 measurements on 120 patients) highlighted the temporal dynamics of SARS-CoV-2 binding antibodies. We estimated the decay of binding reactivity employing a power law mixed model and identified a significant reduction in RBD, S1, and N in all Ig subtypes (Fig. 5b) with half-lives ranging from 67 to 404 days, with IgG N titers decaying the fastest, in line with previous reports[10]. Half-lives of the neutralization relevant IgG responses to RBD and S1 where 125 and 404 days, respectively. Intriguingly, the kinetics of neutralizing antibodies did not mirror the decay rates observed for binding antibodies. Neutralization activity decreased overall at a slower rate, with a half-life of 991 days (Fig. 5c). This was in part due to a mixed reactivity pattern with some individuals showing an increase in neutralization activity post positive SARS-CoV-2 RT-PCR[5], while neutralization activity in others rapidly decayed (Supplementary Fig. 11c).

**Effects of HCoV immunity on SARS-CoV-2 acquisition.** To enable an investigation of interdependencies between pre-existing immunity to HCoV and SARS-CoV-2 infection, we expanded the ABCORA bead antigen array to include S1 proteins of all four circulating HCoVs (HCoV-NL63, HCoV-229E, HCoV-HKU1, HCoV-OC43) (Supplementary Fig. 12a). According to its capacity to monitor antibodies to five coronaviruses we termed the assay ABCORA 5.0 and trained and validated it on the same cohorts as ABCORA 2.0 (Fig. 1). To allow direct comparison with ABCORA 2.0 and use of the neutralization prediction models, we

used the threshold-/SOC-based analysis settings of ABCORA 2.0 also for ABCORA 5.0. Based on ABCORA 5.0 measurements of training cohorts I–III, we devised two random forest-based analysis models. ABCORA 5.4 included solely the 12 SARS-CoV-2 parameters, ABCORA 5.5 included in addition the S1 HCoV measurements adding up to 24 parameters in total. In analogy to ABCORA 2.3, incorporation of HCoV reactivity into the model was advantageous. ABCORA 5.5 provided the highest sensitivity and specificity amongst the analysis algorithms probed ABCORA 5.0 (Supplementary Fig. 12b, Supplementary Table 10).

Interdependencies between antibody reactivity to the four HCoVs and SARS-CoV-2 mirrored what we previously observed for HKU1 with a particular high correlation of IgM reactivity of SARS-CoV-2 and HCoVs in pre-pandemic individuals, particularly in those with recent HCoV infection (Supplementary Fig. 13). HCoV infections are frequent but subject to seasonality and prevalence of individual HCoV infections fluctuates[41,42]. In line with this, the prevalence of HCoV responses measured by ABCORA 5.0 in local blood donors in January 2019 ($N = 285$), May 2019 ($N = 288$), and January 2020 ($N = 252$) varied considerably (Fig. 6). To enable a time-controlled comparison of HCoV reactivity between SARS-CoV-2-infected and healthy donors, we screened blood donors from May 2020 ($N = 672$), when SARS-CoV-2 prevalence was estimated below 2% in Zurich, Switzerland[43], by ABCORA 2.0/5.0 and excluded all samples with SARS-CoV-2 reactivity. The residual May 2020 cohort ($N = 653$) formed a pandemic, healthy donor control group. Interestingly, HCoV reactivity patterns in 2019 and 2020 differed substantially as assessed by one-way ANOVA, with January 2020 showing the comparatively lowest and May 2020 the highest IgA and IgG reactivity, which may indicate a later onset of an HCoV epidemic in 2020 compared to 2019 (t-tests of May 2020 versus other groups shown in Fig. 6).

Most intriguingly, a time-matched analysis comparing May 2020 healthy donors with SARS-CoV-2-positive patients sampled in April, May and June 2020 ($N = 65$) revealed significantly lower HCoV reactivity in SARS-CoV-2 positive patients (Fig. 7a). This pattern was also evident when we extended the analysis to include the full cohort of SARS-CoV-2 infected individuals measured with ABCORA 5.0 ($N = 389$, sampled from March 2020 to February 2021, Supplementary Fig. 14). Overall, these results indicated that pre-existing immune responses to HCoVs may to a certain degree protect against SARS-CoV-2 infection.

**Effects of HCoV immunity in SARS-CoV-2 infection.** To explore interdependencies with HCoV immunity further, we next investigated whether HCoV responses are linked to the evolution of SARS-CoV-2 antibodies. To this end, we analyzed antibody responses in plasma of 204 individuals sampled within 60 days post SARS-CoV-2 diagnosis using a linear regression model adjusted for age, gender, time since positive RT-PCR and HCoV reactivity. To stratify HCoV reactivity into high and low HCoV activity, median logMFI-FOE were defined for each HCoV and antibody class. LogMFI-FOE higher than the corresponding

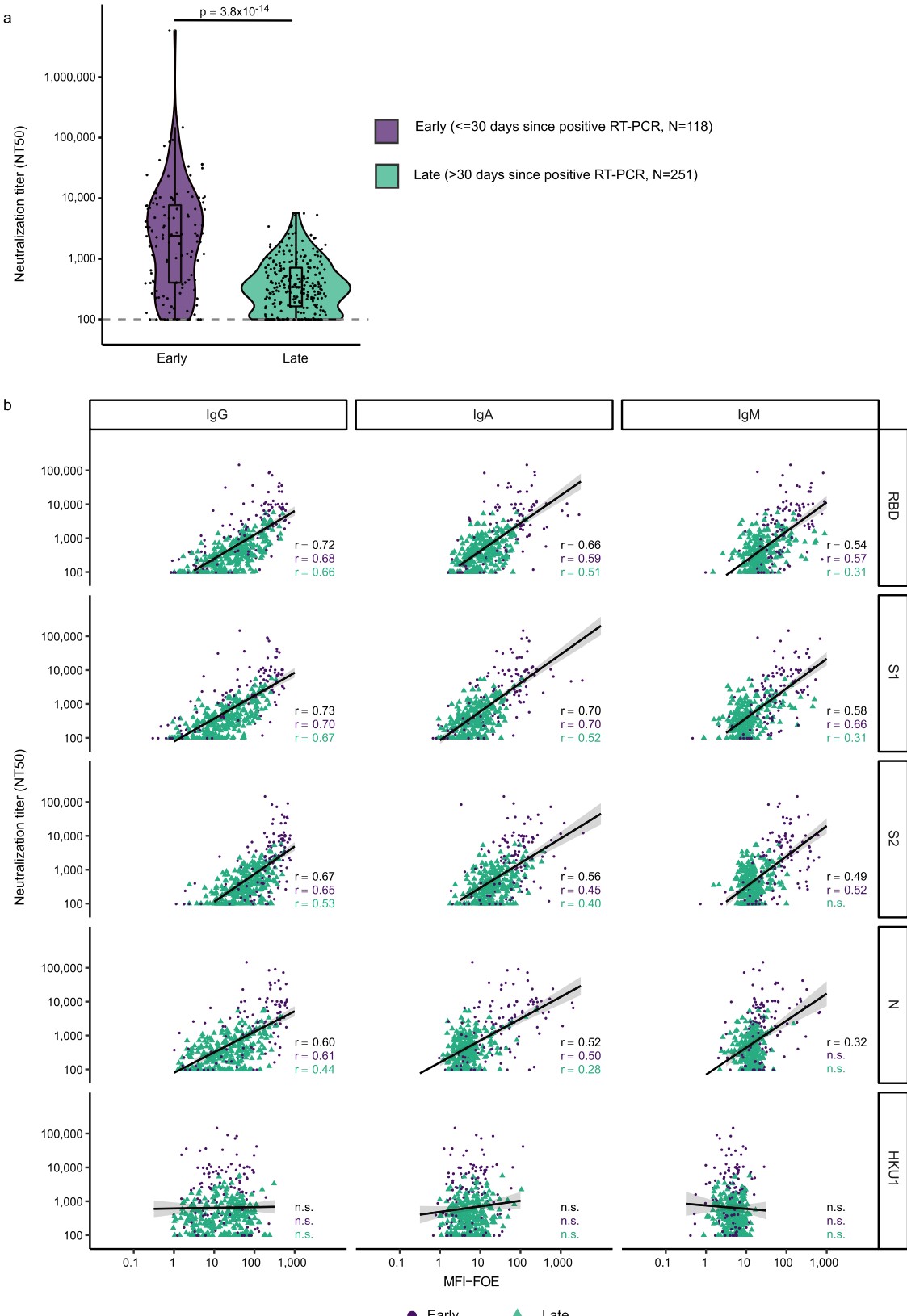

median for at least three HCoVs (HKU1, OC43, NL63 or 229E) in a specific Ig class were ranked as having high HCoV activity within this class. First, only reactivities among the same antibody class were explored in the model (i.e., HCoV IgG high on SARS-CoV-2 IgGs). We observed exceptionally strong inter-dependencies for IgA and IgM responses to SARS-CoV-2, which

all were significantly higher in individuals with high HCoV reactivity (Fig. 7b). This strongly suggests that pre-existing HCoV immunity may provide an advantage in mounting SARS-CoV-2 responses. Interdependencies between HCoV IgG and SARS-CoV-2 specific IgG were only observed for the S2 response. Intriguingly, supporting this finding, HCoV S2 helper responses

**Fig. 3 Association of binding and neutralization activity in early and late infection. a** 50% Neutralization titers (NT50) titers against Wuhan-Hu-1 pseudotype in patients with known positive SARS-CoV-2 RT-PCR date ($N = 369$). Patients were stratified according to time since first diagnosis to investigate early (less than 30 days post RT-PCR, lavender) and late (more than 30 days post RT-PCR, turquoise) neutralization responses. Difference between these two groups was assessed with a linear mixed model with time since RT-PCR (binary variable early/late) as fixed effect and individual as random effect and using Satterthwaite approximation for a two-sided $t$-test on the parameter associated with time since RT-PCR. Boxplots represent the following: median with the middle line, upper and lower quartiles with the box limits, 1.5x interquartile ranges with the whiskers. **b** Linear regression analysis to define association between neutralization (reciprocal NT50) and antibody binding (MFI-FOE). Black lines indicate linear regression predictions and gray shaded areas correspond to the 95% confidence intervals. Results depict early (lavender), late (turquoise) and full cohort (black). n.s. denotes nonsignificant results. Levels of significance are assessed by a two-sided test on the asymptotic $t$ approximation of Spearman's rank correlation, and corrected by the Bonferroni method for multiple testing ($p < 0.05/1200$, see Supplementary Figs. 8b and 9).

were recently found to boost SARS-CoV-2 immunity, in particular S2 antibody activity[31]. To explore if SARS-CoV-2 IgG may build on recent HCoV IgA and IgM responses we next probed whether HCoV IgM and IgA are linked to elevated SARS-CoV-2 specific IgG levels. While no effect was evident for IgM, we observed a significant association of high HCoV IgA activity on all four measured SARS-CoV-2 responses (Fig. 7c, d). This strongly suggests that recent HCoV infection has a beneficial effect on mounting SARS-CoV-2 antibody responses.

In a next analysis, we probed if pre-existing HCoV immunity has an impact on disease severity in COVID-19. To this end, we probed HCoV immunity in 80 hospitalized and nonhospitalized individuals infected for less than 30 days (Fig. 8a). Controlling for age and gender, we found that individuals with high pre-existing HCoV reactivity had significantly lower odds to require hospitalization (logistic regression OR = 0.16, 95% CI (0.04, 0.67), Fig. 8b, Supplementary Fig. 15). A further stratification of patients by whether they required treatment at an ICU showed a lowered likelihood that patients with high HCoV response rates required hospitalization with intensive care (ordinal regression OR = 0.36, 95% CI (0.13, 0.96), Fig. 8b, Supplementary Fig. 15). Thus, individuals with high HCoV levels had a 64% lowered odds of requiring hospitalization according to ordinal rank regression analysis comparing hospitalized in regular wards, in ICU and nonhospitalized individuals (Fig. 8a, b). Collectively, these observations strongly suggest a cross-protective effect of HCoV immunity on shaping the immune defense against SARS-CoV-2.

## Discussion
Definition of SARS-CoV-2 immunity post vaccination and infection is of immediate importance[44–46]. Deciphering antibody correlates of SARS-CoV-2 protection and monitoring vaccine responsiveness are challenging tasks ahead. The magnitude and longevity of protective antibody responses to natural infection and of different vaccines need to be examined to understand parameters that shape protective responses and guide decisions on revaccination in nonresponders and immunization against novel arising SARS-CoV-2 variants[47]. Likewise, creating means to serologically distinguish between de novo infection, reinfection, and vaccine responses, their durability and failures will remain critical for clinical diagnosis.

Here we demonstrate the high utility of multi-parameter seroprofiling in addressing key issues in defining SARS-CoV-2 immunity. Simultaneous detection of antibody responses to a range of SARS-CoV-2 antigens and different Ig classes with ABCORA seroprofiling provided a comprehensive picture of SARS-CoV-2 serologic status in a single examination, which can be useful for clinical diagnosis to determine the presence of reinfection, define reinfection, and respond to vaccination. Computational modeling also allowed predicting plasma neutralization capacity from ABCORA results, enabling a comprehensive assessment of SARS-CoV-2 antibody dynamics and their interplay with HCoV responses. We studied two ABCORA assay

versions that both measured HCoV reactivity alongside the 12 SARS-CoV-2 parameters. ABCORA 2 included the S1 antigen of HKU1. ABCORA 5 included S1 of all four circulating HCoVs. Notably, computational models that included the HCoV measurements allowed a higher precision in determining SARS-CoV-2 seropositivity, highlighting interdependencies between HCoV and SARS-CoV-2 responses that need to be resolved.

Recording reactivity against all four HCoVs in SARS-CoV-2 uninfected and infected individuals we observed intriguing associations. Uninfected individuals displayed higher HCoV reactivity compared to infected individuals suggesting a contribution of HCoV immunity to early defense against SARS-CoV-2. HCoV immunity may also have positive effects in SARS-CoV-2 infection. In agreement with other reports, we noted a cross-feeding of SARS-CoV-2 and HCoV responses in SARS-CoV-2 infection[31,48] with individuals with high HCoV reactivity developing higher SARS-CoV-2 antibody levels. Most notably, pre-existing HCoV immunity had an impact on disease severity in our cohort. SARS-CoV-2 infected individuals with low HCoV reactivity had a higher likelihood of requiring hospitalization.

While our study solely measured antibody responses, a potential protective effect of HCoV immunity against SARS-CoV-2 acquisition should not be viewed restricted to antibody activity. Antibodies and cellular immunity may both be relevant and act in concert[31,49–51]. Alternatively, antibody responses measured in the present study may solely document recent HCoV infection and deliver a surrogate measurement of other protective HCoV responses. The link between higher HCoV and SARS-CoV-2 reactivity in infected individuals is particularly intriguing. Strongest effects were seen for IgM and IgA HCoV responses, suggesting that recent HCoV immunity provides an early boost to SARS-CoV-2 antibody development. Whether this is due to cross-reactive B cell responses on which the SARS-CoV-2 immunity builds on and matures or whether cross-reactive T helper activities play a dominant role as suggested[31] will be important to resolve in forth-coming studies. The exact role and timing of HCoV responses influencing SARS-CoV-2 antibody responses also remain to be defined. Early low-affinity HCoV responses may have a positive impact on the development of SARS-CoV-2 immunity by forming an immune memory on which to build, whereas amplification of non-protective cross-reactive HCoV antibodies according to the antigenic sin principle may have negative effects[30].

Although association studies such as ours cannot formally define causality, the implications of our findings are evident: Prior immunity to HCoV may protect to some extent against SARS-CoV-2 acquisition, may provide a boost to the development of SARS-CoV-2 specific immunity and with this lower the risk for severe hospitalization. A modest protective effect by HCoV immunity would be a plausible explanation for the high proportion of asymptomatic and mild SARS-CoV-2 infections[52,53]. Even more intriguing are future perspectives. As others and we have shown, SARS-CoV-2 and HCoV immunity to infection is

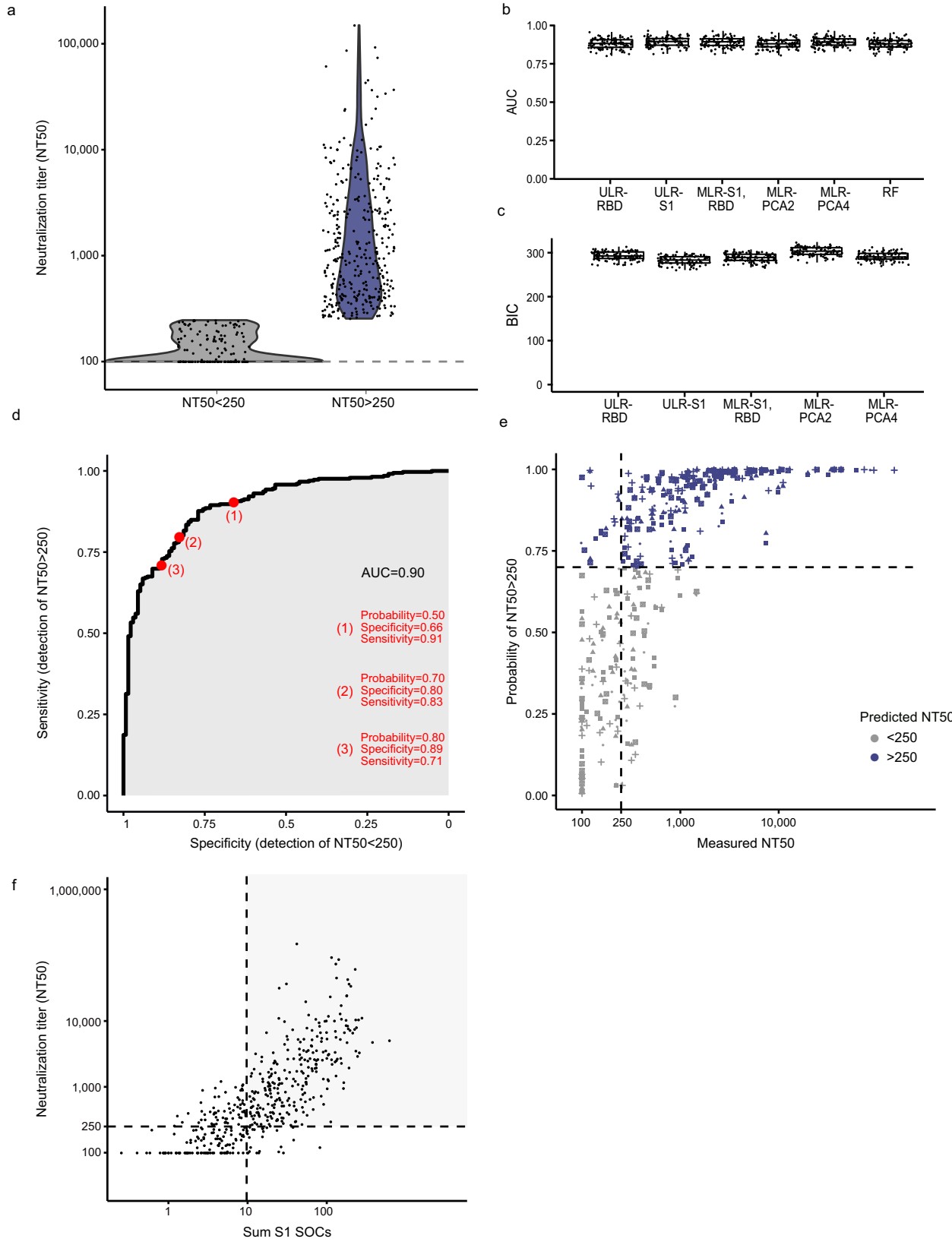

often not long-lasting (Fig. 5, Supplementary Fig. 11)[5,54], a limitation that SARS-CoV-2 vaccines hope to overcome. Should SARS-CoV-2 responses in turn provide a degree of defense against HCoV infection, broad protection against coronaviruses may be in reach.

## Methods

**Human specimen**. Serum and plasma samples collected pre and post emergence of SARS-CoV-2 in Switzerland (pre and post February 2020, respectively) were included. No patient enrollment was conducted for the present study. All experiments involving samples from human donors were conducted with the approval of the responsible local ethics committee (Kantonale Ethikkommission) Zurich,

**Fig. 4 Predicting neutralization capacity as a function of binding activity. a** SARS-CoV-2 positive donors ($N = 467$) were stratified into high neutralizers (NT50 > 250, $N = 332$; blue) and no/low neutralizers (NT50 < 250, $N = 135$; gray), based on their neutralization activity against Wuhan-Hu-1. **b**, **c** Comparison of the prediction ability of six different classification models using 100 cross-validation sets (divided as 80% for training and 20% for validation). **b** Comparison of models by area under the curve (AUC). Each dot corresponds to one cross-validation set. **c** Bayesian information criterion (BIC) of the five models based on logistic regression. The different models are: Univariable logistic regressions (ULR). ULR-RBD: mean of MFI-FOE RBD. ULR-S1: mean of MFI-FOE S1. Multivariable logistic regression (MLR). MLR-S1, RBD: mean of S1 reactivity and mean of RBD reactivity. MLR-PCA2 and MLR-PCA4: MLR of 2 and 4 first axis of PCA analysis, respectively. PCA was based on all 12 SARS-CoV-2 antibody reactivities measured by ABCORA 2.0. Random forest (RF) including all antibody reactivities measured by ABCORA 2.0. Boxplots represent the following: median with the middle line, upper and lower quartiles with the box limits, 1.5x interquartile ranges with the whiskers and outliers with points. **d** ULR-S1 estimated ROC curve based on full data set ($N = 467$). **e** Measured NT50 value versus probability of NT50 > 250 as predicted by ULR-S1 in five randomly chosen validation sets (each symbol corresponds to a validation set). Purple colored symbols indicate a higher than 0.70 probability of the respective sample to be neutralizing at NT50 > 250 and are therefore denoted as high neutralizers. Gray indicates samples with predicted neutralization NT50 < 250, therefore classified as no/low neutralizers. **f** Neutralization prediction based on a modified ULR-S1 model utilizing the diagnostic readout SOC instead of MFI-FOE values as input. Measured NT50 value versus sum of S1 SOC values (IgG, IgA, IgM) are depicted. Dashed lines correspond to a NT50 = 250 horizontally and the sum S1 SOCs = 9.7 vertically. The sum S1 SOCs = 9.7 corresponds to the thresholds depicted for ULR-S1 in (**d**, **e**). The gray shaded area corresponds to true positives (individuals with NT50 > 250 predicted as high neutralizers).

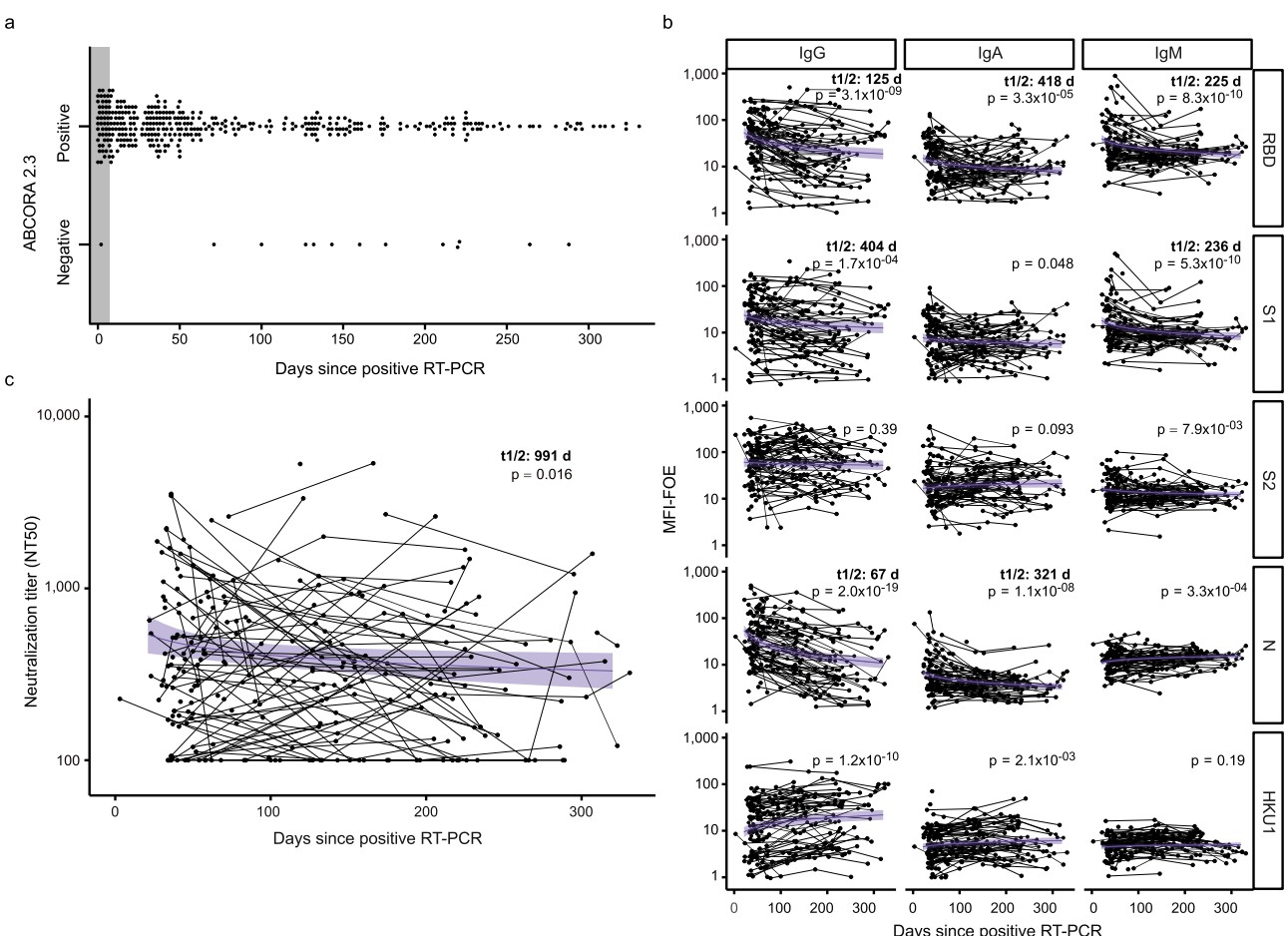

**Fig. 5 Monitoring temporal evolution of antibody responses. a** ABCORA 2.3 definition of seropositivity in donors with positive RT-PCR confirmed SARS-CoV-2 infection and known RT-PCR date ($N = 369$). Seropositivity rating in relation to plasma sampling time point post diagnosis is depicted. Gray shaded area highlights the first seven days since positive RT-PCR detection. **b** Power law model, with time since RT-PCR as fixed effect and individual as random effect, estimating the decay of antibody binding activity based on ABCORA 2.0 measurements at 1–4 longitudinal time points in 120 individuals totaling in 251 measurements. Purple lines correspond to the models estimation and purple shaded areas to the 95% confidence intervals. Antibody half-lives (t1/2 in days) from significant models are depicted. Significance was assessed using Satterthwaite approximation for a two-sided t-test on the slope parameters. **c** Power law model, with time since RT-PCR as fixed effect and individual as random effect, estimating the decay of neutralizing capacity on 251 measurements from 120 individuals. Only individuals with NT50 > 100 at their first measurement were used to estimate the half-life. The purple line corresponds to the model estimation and the purple shaded area to the 95% confidence intervals. Significance was assessed using Satterthwaite approximation for a two-sided t-test on the slope parameters.

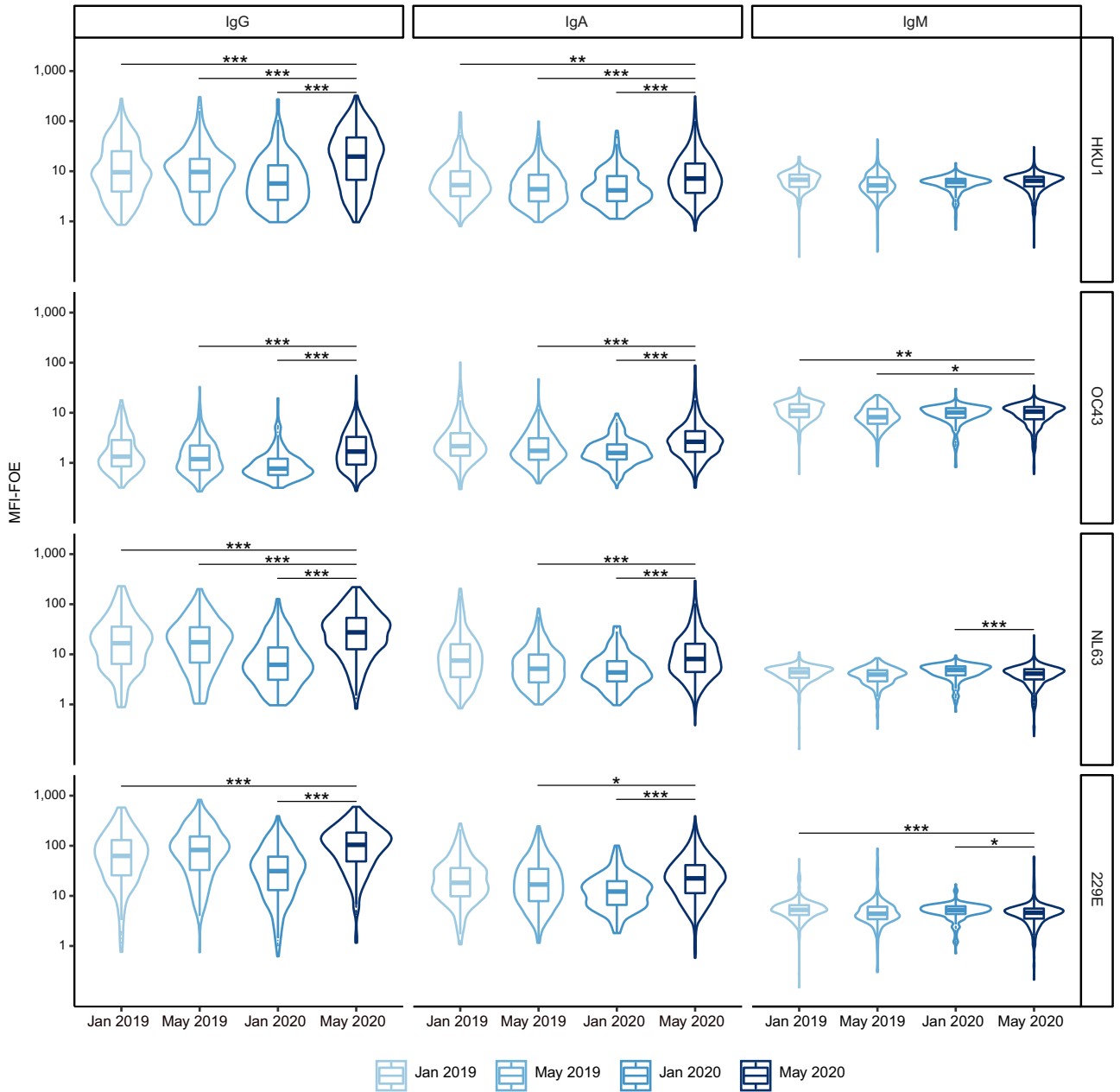

**Fig. 6 Seasonal and annual fluctuation in HCoV reactivity.** Reactivity to human coronaviruses (HCoV-NL63, HCoV-229E, HCoV-HKU1, HCoV-OC43) was compared by ABCORA 5.0. Reactivity in healthy blood donors from 2019 and 2020 was compared. Pre-pandemic samples included: January 2019 ($N = 285$), May 2019 ($N = 288$), January 2020 ($N = 252$). Samples from May 2020 ($N = 672$) were collected during the pandemic in Switzerland. Only samples without SARS-CoV-2 specific reactivity as defined by ABCORA were included ($N = 653$). Stars correspond to levels of significance of two-sided t-tests comparing the indicated groups. Levels of significance are corrected by the Bonferroni method for multiple testing and indicated as follows: *$p < 0.05/36$, **$p < 0.01/36$, ***$p < 0.001/36$. Boxplots represent the following: median with the middle line, upper and lower quartiles with the box limits, 1.5x interquartile ranges with the whiskers and outliers with points.

Switzerland (BASEC Nrs 2020-01327, 2020-00363; 2021-00437; 2020-00787), in accordance with the provisions of the Declaration of Helsinki and the Good Clinical Practice guidelines of the International Conference on Harmonization. Samples were obtained from the following sources: (i) Zurich blood donation services (ZHBDS): Anonymized healthy adult plasma from pre-pandemic time points (January 2019, May 2019 and January 2020) and from the first wave of the pandemic in Zurich, Switzerland (May 2020) were provided by the ZHBDS internal serum repository and consent for this study was waived by the ethics committee (BASEC 2021-00437). (ii) Anonymized leftover specimens from routine diagnostics at the Institute of Medical Virology, University of Zurich, the University Children Hospital Zurich and the Cantonal Hospital Winterthur (BASEC Nrs 2020-01327, 2021-00437). Written informed consent was obtained from all participants whose sample was taken during the pandemic at the University Hospital Zurich (BASEC 2020-01327). For pandemic samples from other hospitals and pre-

pandemic samples consent was waived by the ethics committee. (iii) Healthcare workers with RT-PCR confirmed SARS-CoV-2 infection participating in a study at the University Hospital Zurich (BASEC 2020-00363). Written informed consent was obtained from all participants. (iv) Male plasma donors participating in a SARS-CoV-2 plasma therapy study conducted at the University Hospital Zurich (CPT-ZHP, Swissmedic 2020TpP1004; BASEC 2020-00787). Written informed consent for research was obtained from all participants. The reporting of all human and patient data is in compliance with STROBE statement. Pre-pandemic (SARS-CoV-2 negative, $N = 825$) and confirmed SARS-CoV-2 positive samples ($N = 389$) were divided into training and validation cohorts (Supplementary Table 3). Available demographics data on gender, age, time since positive RT-PCR and symptom onset, and hospitalization status are reported in Supplementary Table 11. The SARS-CoV-2 training cohort ($N = 175$) included plasma collected during infection ($N = 114$) and convalescence ($N = 61$). Per donor only one sampling time

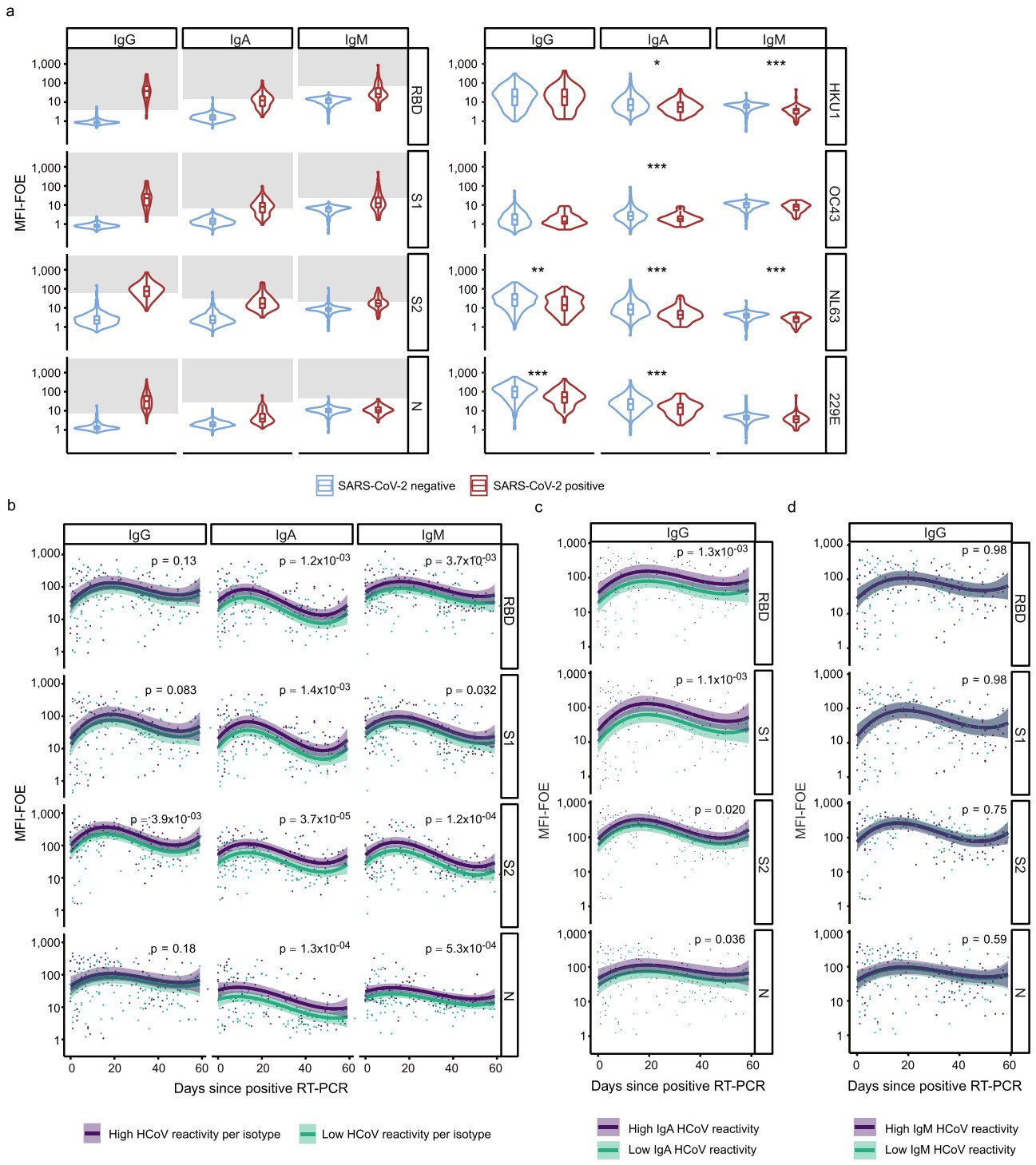

point was included, longitudinal samples of donors included in the training cohort were not included in the validation cohort to ascertain independence when assessing the sensitivity and specificity of the different diagnostic methods. The SARS-CoV-2 validation cohort ($N = 214$), comprised plasma collected during infection ($N = 90$, one sampling time point per donor) and convalescence ($N = 124$, 79 convalescent patients with 1–4 longitudinal samples). Multiple time points of convalescent patients were included in the validation data set to capture a wide spectrum of waning antibody titers. Cross-sectional analysis was based on samples with known time since positive RT-PCR ($N = 369$) or known time since symptom onset ($N = 333$), both including the longitudinal analysis observations. Both, time since positive RT-PCR and time since onset were known for 330 samples, with a median time of three days between symptom onset and RT-PCR (1st–3rd quartile: 1–7 days). Longitudinal analysis of antibody reactivity was based on 251 observations from 120 convalescent patient with known time since positive RT-PCR and time since symptom onset. Neutralization was measured on 467

SARS-CoV-2 RT-PCR positive samples ($N = 369$ with known time since positive RT-PCR, $N = 333$ with known time since symptom onset).

We further evaluated cross-reactivity ABCORA 2.0 and 5.0 in left-over plasma from routine diagnostics in a prepandemic control group with documented, recent HCoV infection (Training II, $N = 75$; OC43 ($N = 27$), HKU1 ($N = 17$), NL63 ($N = 22$), 229E ($N = 9$)). Circulating HCoV are commonly only screened for in hospitalized, severe respiratory infections and immune compromised individuals who routinely undergo a broad screening for respiratory infections. Hence, in this patient group both reduced antibody reactivity due to immune compromising or elevated HCoV antibody reactivity due to recent or recurring HCoV infection may occur. As this group is diagnostically relevant we considered it prudent to include this cohort as Training II data set to verify if cross-reactivity with SARS-CoV-2 in ABCORA occurs in this setting. Training II data were not included in the threshold definition to not over-represent individuals with severe illness. This HCoV infected group displayed overall lower reactivity with SARS-CoV-2 than plasma from

**Fig. 7 Effects of pre-existing HCoV immunity during SARS-CoV-2 acquisition. a** Time-matched comparison of ABCORA 5.0 reactivity for SARS-CoV-2 and HCoVs in healthy and SARS-CoV-2 infected individuals. Healthy donors were sampled in May 2020 ($N = 653$; blue). Plasma from SARS-CoV-2 infected individuals were collected between April–June 2020 ($N = 65$; red). See Supplementary Fig. 14 for analysis on the full SARS-CoV-2 positive cohort ($N = 389$). Gray boxes indicate values above the individual MFI-FOE cut-offs for SARS-CoV-2 specific responses for each antigen. Stars correspond to levels of significance of two-sided $t$-tests comparing negative versus positive patients. Levels of significance are corrected by the Bonferroni method for multiple testing and indicated as follows: *$p < 0.05/12$, **$p < 0.01/12$, ***$p < 0.001/12$ (IgG HKU1: $p = 0.66$, IgG OC43: $p = 0.45$, IgG NL63: $p = 3.3 \times 10^{-04}$, IgG 229E: $p = 1.6 \times 10^{-05}$, IgA HKU1: $p = 1.8 \times 10^{-03}$, IgA OC43: $p = 1.3 \times 10^{-05}$, IgA NL63: $p = 1.4 \times 10^{-07}$, IgA 229E: $p = 3.0 \times 10^{-05}$, IgM HKU1: $p = 3.3 \times 10^{-08}$, IgM OC43: $p = 4.3 \times 10^{-03}$, IgM NL63: $p = 1.1 \times 10^{-07}$, IgM 229E: $p = 2.7 \times 10^{-02}$). Boxplots represent the following: median with the middle line, upper and lower quartiles with the box limits, 1.5x interquartile ranges with the whiskers and outliers with points. **b** Linear regression models showing the association between SARS-CoV-2 and HCoV signals in 204 SARS-CoV-2 positive patients with known dates of first positive RT-PCR detection. Influences within the same antibody class are investigated. The models were adjusted on age (spline with 3 degrees of freedom), gender, time since positive RT-PCR (spline with 3 degrees of freedom) and level of HCoV reactivity. Samples are defined to harbor high HCoV reactivity if they show ABCORA 5.0 HCoV logMFI-FOE values higher than the corresponding median in at least 3 HCoV measurements (HKU1, OC43, NL63 or 229E). Curves correspond to the models estimation and shaded areas to the 95% confidence intervals. $p$-values were obtained by running a two-sided Student $t$-test on the parameter associated to HCoV reactivity in the linear regression. **c** Linear regression model showing the association between SARS-CoV-2 IgG and HCoV IgA signals. Curves correspond to the models estimation and shaded areas to the 95% confidence intervals. $p$-values were obtained by running a two-sided Student $t$-test on the parameter associated to HCoV reactivity in the linear regression. **d** Linear regression model showing the association between SARS-CoV-2 IgG and HCoV IgM signals. Curves correspond to the models estimation and shaded areas to the 95% confidence intervals. $p$-values were obtained by running a two-sided Student $t$-test on the parameter associated to HCoV reactivity in the linear regression.

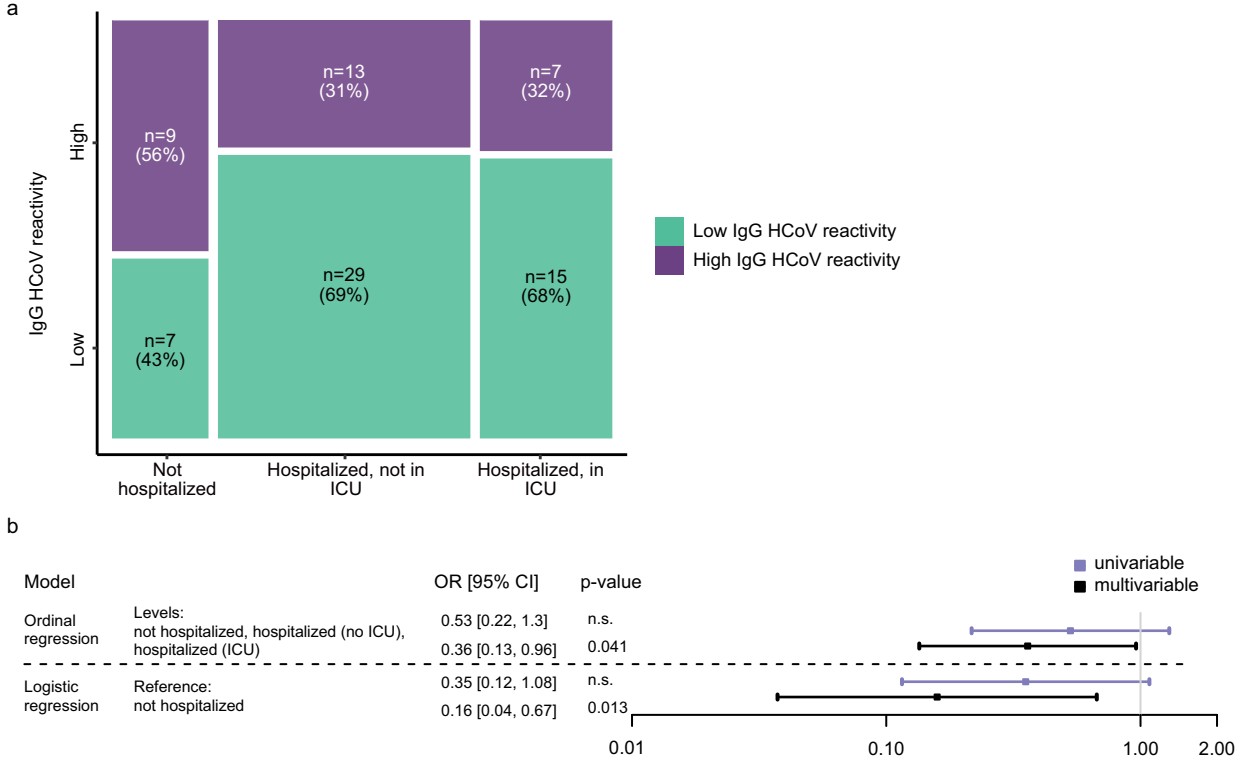

**Fig. 8 Impact of HCoV immunity on COVID-19 severity. a** Association of hospitalization status (not hospitalized ($N = 16$); hospitalized not in ICU ($N = 42$); hospitalized in ICU ($N = 22$)) and high or low IgG HCoV reactivity. Rectangle sizes correspond to the proportion of included patients. **b** Influence of HCoV reactivity (low/high) on the hospitalization status in a subset of $N = 80$ patients, as estimated with odds ratios, in an ordinal regression (with levels = not hospitalized ($N = 16$); hospitalized not in ICU ($N = 42$); hospitalized in ICU ($N = 22$)) and a logistic regression (reference = not hospitalized ($N = 16$); versus all hospitalized ($N = 64$)). Data is presented as parameter estimation and its 95% confidence interval. Level of significance of the parameter is obtained with a two-sided $t$-test ($p$-value is displayed if <0.05, otherwise indicated as n.s.).

healthy adults but importantly showed no indication of cross-reactivity (Fig. 1a, b). Pandemic samples from anonymous blood donors with unknown SARS-CoV-2 status collected in May 2020 ($N = 672$) were not included in training and validation cohorts.

**Reagents and cell lines**. His-tagged SARS-CoV-2-derived antigens (receptor binding domain (RBD), subunit S1 (S1), subunit S2 (S2), nucleocapsid protein (N)) and S1 of the four circulating HCoVs (HKU1, OC43, NL63, 229E) were purchased from Sino Biological Europe GmbH, Eschborn, Germany (Supplementary

Table 12). Sources, specifics and concentration of detection and control antibodies and sera used for ABCORA and neutralization tests are listed in Supplementary Table 13. 293-T cells were obtained from the American Type Culture Collection (ATCC CRL-11268)[55]. HT1080/ACE2cl.14 cells[33] were kindly provided by P. Bieniasz, Rockefeller University, NY. Both cell lines were cultured in DMEM containing 10% FCS.

**Design of multiplex bead assay ABCORA 2.0**. We established two bead-based multiplexed SARS-CoV-2 immunoassays (ABCORA 2.0 and ABCORA 5.0) that

included a range of SARS-CoV-2 and HCoV antigens (Sino Biological Europe GmbH, Eschborn, Germany, Supplementary Table 12). Four SARS-CoV-2 antigens - RBD, S1, S2 and N - were included in both ABCORA 2.0 and ABCORA 5.0. ABCORA 2.0 included in addition S1 of HCoV-HKU1, ABCORA 5.0 included S1 of all circulating HCoVs (HCoV-NL63, HCoV-229E, HCoV-HKU1, HCoV-OC43). In brief, individual MagPlex beads (Luminex Corporation, Austin, TX) with unique fluorescent bead regions were chosen for each antigen, beads were coupled and mixtures of antigen-coupled beads incubated with patient plasma in a 96-well plate set-up. Median Fluorescent Intensity (MFI) of bead-bound plasma antibodies were measured utilizing a FlexMap 3D reader (Luminex Corporation, Austin, TX). We designed the assay to fulfill the following criteria: (i) high specificity, sensitivity and reproducibility, (ii) flexible multiplex design that allows straightforward addition and/or alteration of antigens; (iii) wide dynamic range; (iv) optional quantification of antibody responses; (v) optional recording of antibody responses to HCoVs and (vi) use in routine diagnostics and research.

We chose a sterically orientation capture via anti-His antibodies to ensure a homogenous antigen display. Therefore, carboxylated MagPlex beads (Luminex Corporation, Austin, TX) were coupled with anti-His antibody (Sino Biological Europe GmbH, Eschborn, Germany, Supplementary Table 13) and then coupled with His-tagged antigens using Bio-Plex Amine coupling (Bio-Rad Laboratories AG, Cressier, Switzerland) according to the manufacturer's instructions and as described[56].

Serum/plasma titration is in general considered the most accurate strategy to retrieve quantitative information on antibody reactivity. However, in diagnostic use tests ideally should deliver (semi)-quantitative information from a single serum dilution to permit a sufficient throughput. The finalized assay conditions covered a 2-log MFI range across all probed antigen-Ig combinations (Supplementary Fig. 1). Ratifying the validity of using a single plasma dilution, we confirmed that plasma from SARS-CoV-2 positive patients and pre-pandemic SARS-CoV-2 negative plasma samples show optimal dose response curves over a wide plasma dilution range (Supplementary Fig. 1f). Importantly, a 1/100 dilution of plasma was in all cases close to the maximum signal, underlining that increasing plasma concentration would not increase signal intensity but rather endanger decreasing signals due to prozone effects (Supplementary Fig. 1f).

Maximal anti-His antibody loading was achieved at 5 μg antibody per million beads (Supplementary Fig. 1c) and used as standard coupling condition. In the final protocol, five million anti-His antibody coupled magnetic beads were incubated with His-tagged antigens diluted in PBS at a concentration of 320 nM. Phycoerythrin (PE)-labeled secondary antibodies specific to IgG, IgA or IgM were used as detector antibodies (Supplementary Table 13). Quality control of the antigen loading was performed by incubating the beads with monoclonal antibodies targeting the corresponding CoV-derived antigen as detailed in Supplementary Table 12. Analysis was performed with the FlexMap 3D reader (Luminex Corporation, Austin, TX) with the acquisition of at least of 50 beads per bead region. Results are recorded as MFI per bead region.

Several control measures were installed to ascertain inter- and intra-assay performance. To ascertain a low assay-to-assay variability, large batches of individual antigen-loaded beads were prepared and frozen in aliquots until use at -20 °C to circumvent decay of the antigen-coupled beads (Supplementary Fig. 1). Individual coupled beads were mixed on the test day to yield the required antigen bead cocktail. Cocktails contained 60 beads per bead region per μl. In addition to the SARS-CoV-2 and HCoV bead regions, each cocktail included an empty bead region (no antigen coupled) to control for unspecific binding. Quality control and validation procedures for the FlexMap 3D instrument were done on each day of experiment according to manufacturer's instructions. The variability of the assay was analyzed as follows[56]: plasma samples from 20 RT-PCR confirmed SARS-CoV-2 infected patients were pooled and tested over a range of seven dilutions in 31 different titrations performed on 10 different days (Supplementary Fig. 2). Across all antigens and Ig classes, signals were retained over the test period of 25 days post bead coupling. Coefficients of variation (CV) of the binding signal across titrations of the 12 antigen-Ig class combinations proved low (range: 0.010–0.128, median 0.059, Supplementary Fig. 3c, d, Supplementary Tables 1, 2). Same-day and day-to day variability proved low and acceptable (Supplementary Fig. 3e, f). Below a 1/100 plasma concentration, CV increased markedly (Supplementary Fig. 3d), defining 1/100 as highest concentration (lowest plasma dilution) to be tested in the assay. A 1/100 plasma dilution was thus defined as the basic dilution for screening plasma in ABCORA 2.0 when a qualitative (i.e., presence or absence of SARS-CoV-2 specific antibodies) or semi-quantitative (i.e., MFI signal intensity) readout is required.

All ABCORA measurements were derived from single measurements unless stated otherwise. To measure SARS-CoV-2 specific antibodies in patient plasma, heat inactivated plasma (1 h at 56 °C) was diluted 1/100 in PBS-BSA 1% unless otherwise stated. 50 μl diluted plasma were incubated with 50 μl of the ABCORA antigen bead cocktail for 30 minutes at room temperature in 96-well plates, washed three times with PBS-BSA 1% and incubated in separate reactions with phycoerythrin (PE)-labeled detector antibodies for IgG, IgA or IgM at a final concentration of 1/500 in PBS-BSA 1%. This dilution was previously defined by titration of the detector antibodies to yield optimal MFI signals. After 45 minutes of incubation at room temperature, beads were washed three times with PBS-BSA 1% and analyzed in 96-well plates on the FlexMap 3D reader (Luminex Corporation, Austin, TX). A minimum of 50 bead reads per antigen was acquired.

To control for genuine cross-reactive antibodies, each plasma sample was assessed with beads without antigen (empty bead control) in combination which each detector antibody. For analysis, raw MFI values were transformed to MFI-FOE to correct for background binding. We established mean empty bead MFI-FOE for IgG, IgA and IgM of pre-pandemic healthy donors ($N = 1016$) and set the mean MFI-FOE + 4x standard deviation as threshold for the empty bead control. In absolute levels, these thresholds amounted to MFI-FOE 41.58 (IgG), 55.91 (IgA) and 269.47 (IgM). Measurements for which the empty bead control recorded values above this threshold were considered invalid and repeated.

Each Luminex analysis 96-well plate was set up to contain the same set of control samples, namely 7 serial 4-fold dilutions of a SARS-CoV-2 positive control donor pool ($N = 20$ donors, starting dilution 1/100, Supplementary Fig. 1f) and a pre-pandemic healthy donor pool (dilution 1/100, $N = 20$ donors, Supplementary Fig. 1f). These positive and negative controls allow to control assay performance across independent measurements and in addition enable retrospective standardization against external controls if needed.

**Definition of SARS-CoV-2 seropositivity in ABCORA 2.0 and ABCORA 5.0**. To distinguish SARS-CoV-2-specific from cross-reactive antibodies, we defined MFI-FOE thresholds for each of the 12 SARS-CoV-2 antigen and Ig class combinations based on plasma antibody reactivity in training cohorts (Supplementary Tables 3 and 4). These included pre-pandemic healthy donor plasma (Training I, $N = 573$), donors with recent HCoV infection (Training II, $N = 75$; OC43 ($N = 27$), HKU1 ($N = 17$), NL63 ($N = 22$), 229E ($N = 9$)) and donors with confirmed SARS-CoV-2 infection (Training III, $N = 175$). Thresholds were set to minimize false-positives while ensuring sensitivity for SARS-CoV-2 antibody detection and to reach an overall specificity above 99% and included levels for border-line reactivity for IgG RBD, IgG S1 and IgG N to allow also modest antibody reactivity to these antigens to be examined. MFI-FOE reads of individual samples were transformed into signal-over cut-off (SOC) values (MFI-FOE/threshold). SOC values are used for assessing positive reactivity for each individual antigen-antibody class combination, with SOC > 1 denoting positive reactivity, SOC < 1 denoting negative reactivity. When setting individual thresholds, it must be considered that for each of the 12 probed activities cross-reactivities may occur. With 12 individual SOC parameters recorded, overall specificity will decrease if any positive SOC independently suffices to rate a sample overall as SARS-CoV-2 antibody positive. To exemplify: Assuming independent responses, even a high 99% specificity for each antigen will add up to an overall low 88% specificity across the entire assay. We thus required for SARS-CoV-2 positive calling in ABCORA 2.0 a minimum of two specificities to reach activity above threshold. The combined SOC values used to define the overall serostatus of a given sample are detailed in Supplementary Table 5. For IgG RBD, S1 and N, for which we also recorded border-line SOC activity, we allowed for a combination of 1 antigen reactivity SOC > 1, the second reactivity SOC > border line. The final threshold and positive call criteria allowed for a differentiation of partial (early seroconversion with only IgM and IgA responses) to full seroconversion (including IgG responses) (Supplementary Table 5). In addition, the criteria denote samples with weak reactivity and/or indeterminate reactivity (Supplementary Table 5).

To ease comparison between ABCORA 2.0 and ABCORA 5.0 the same threshold cut-offs were used for ABCORA 5.0. We chose not to create specific cut-off thresholds for HCoV antibody reactivity as an accurate definition of a negative response is complex due to the wide-spread exposure to HCoVs and considerable antibody cross-reactivity between them. HCoV responses were however included in the statistical analyses as MFI-FOE values.

**Definition of SARS-CoV-2 seropositivity using logistic regression classification**. Classification of seropositive versus seronegative samples in ABCORA 2.1 was realized using logistic regression. The identical training and validation data used for the establishment for ABCORA 2.0 were used. As the ABCORA 2.0 binding reactivities were highly correlated, we included the following variables in the model (Supplementary Fig. 6a): the mean value of all IgG MFI-FOE responses (RBD, S1, S2, N), the mean value of the IgA MFI-FOE responses against RBD, S1 and S2, and the mean value of the IgM MFI FOE responses against RBD, S1 and S2. IgA and IgM responses to N were excluded as they were not clustering with the other responses of the same Ig class (Supplementary Fig. 6a). The logistic regression was used to estimate and predict the probability of a given sample to be positive (p) as described in equation (1).

$$p = \frac{\exp\left(\begin{array}{c}\beta 0 + \beta G * \mathrm{mean}(IgG : RBD, S1, S2, N) + \beta A * \mathrm{mean}(IgA : RBD, S1, S2) \\ + \beta M * \mathrm{mean}(IgM : RBD, S1, S2)\end{array}\right)}{1 + \exp\left(\begin{array}{c}\beta 0 + \beta G * \mathrm{mean}(IgG : RBD, S1, S2, N) + \beta A * \mathrm{mean}(IgA : RBD, S1, S2) \\ + \beta M * \mathrm{mean}(IgM : RBD, S1, S2)\end{array}\right)}$$

(1)

Parameters $\beta 0$, $\beta G$, $\beta A$ and $\beta M$ were estimed on the training dataset. A sample was then defined as positive if its predicted probability of being positive was above a threshold $c'$. This threshold was defined as to obtain a specificity of at least 0.99 and maximal sensitivity on the training dataset (similarly to c for the random forest). In summary, in ABCORA 2.1, any new sample is defined as seropositive if its probability of being seropositive as estimated by the logistic regression is above $c'$. Analyses were performed in R version 3.6.3.

**Definition of SARS-CoV-2 seropositivity using random forest classification**. Classification of seropositive versus seronegative samples in context of ABCORA 2.0 and ABCORA 5.0 was realized using a random forest approach following the basic setup of random forests as described in[57]. The random forest itself was built of an ensemble of 1000 classification trees using MFI-FOE responses (IgA, IgG and IgM against RBD, S1, S2, N). The probability of a sample being positive as predicted by the random forest is the average of the probabilities over all 1000 trees. Finally, a sample is defined as positive if its probability of being positive is above a threshold c, which is defined as to obtain a specificity of at least 0.99 and a maximal sensitivity on the training dataset. In summary, any new sample is defined as seropositive if its probability of being seropositive as estimated by the random forest is above the threshold c. We conducted a series of random forest analyses that considered either only SARS-CoV-2 responses or SARS-CoV-2 and HCoV responses in ABCORA 2.0 and ABCORA 5.0: ABCORA 2.2 and ABCORA 2.3 were trained and used for prediction on ABCORA 2.0 data and included only SARS-CoV-2 responses or SARS-CoV-2 and HKU1 responses, respectively. ABCORA 5.4 (SARS-CoV-2 responses only) and ABCORA 5.5 (SARS-CoV-2 and HCoV responses) were trained on ABCORA 5.0 data. Details on the data inclusion for the respective models are listed in Supplementary Table 3 and Supplementary Table 10. Analyses were performed in R version 3.6.3 using packages randomForest and ranger[58–61].

To ensure robustness of our findings, we performed a sensitivity analysis by randomizing the training and validation datasets using a 5-fold cross validation method. Both sets of positives and negatives samples were divided in five equal parts and we defined that way five validation sets (consisting of the i-th set of positives and the i-th set of negatives, $i = 1.5$). The rest of the data was used for training the ABCORA 2.3 random forest. For each validation set, the specificity and sensitivity of the random forest were computed on the training, validation and training+validation sets (Supplementary Table 6).

**Validation and verification using external controls**. We used the Anti-SARS-CoV-2 Verification Panel for Serology Assays (NIBSC code: 20/B770, NIBSC) to verify the performance of the ABCORA 2.0 and ABCORA 2.3 test. Serum samples of the verification panel measured by ABCORA 2.0/2.3 as described and results compared with the results of commercially available assays reported by the NIBSC ([32] and Supplementary Table 7). We further verified the sensitivity of the ABCORA 2 test in detecting SARS-CoV-2 infection in a direct comparison with commercial tests. Antibody status of plasma from HKU1 positive individuals ($N = 171$) were analyzed with the following test systems: Included test systems targeted the N protein (Elecsys® Anti-SARS-CoV-2 (Roche Diagnostics GmbH)), the RBD region of the S protein (Elecsys® Anti-SARS-CoV-2 S assay (Roche Diagnostics GmbH)), and the S1 subunit (EUROIMMUN Anti-SARS-CoV-2 ELISA (IgG)) (Supplementary Table 8). All assays were performed according to the manufacturer's instructions in the diagnostics unit of the Institute of Medical Virology, University of Zurich, Switzerland.

**SARS-CoV-2 binding antibody titers**. To define binding antibody titers, eight serial 4-fold dilutions starting with a 1/25 dilution of plasma were prepared and measured in ABCORA 2.0. To derive quantitative information, MFI values were corrected for background activity (MFI-empty bead control) and we defined the area under the MFI curve (AUC) across the dilution series for each antigen-Ig combination. As a second quantitative readout, we calculated 50% effective titer concentrations (EC50) using a four-parameter logistic curve (y = Bottom + (Top-Bottom)/(1 + 10^((logEC50-X) ∗ HillSlope))).

**Quantification of SARS-CoV-2 S1 and RBD activity**. We used two approaches to standardize SARS-CoV-2 S1 and RBD activity. The first was based on the S1/RBD specific antibody CR3022 ([38] and Supplementary Table 13). Serial dilutions of IgG, IgA and IgM versions of CR3022 were used to create standard curves on RBD and S1 coated beads. The linear range of the standard and sample dilution curve was used for quantitation. We fitted a four-parameter logistic curve (y = Bottom + (Top-Bottom)/(1 + 10^((logEC50-X) ∗ HillSlope))) (Supplementary Fig. 7b) through which MFI values of measured samples are interpolated into a corresponding concentration of antibody (μg/ml). We used this approach to quantify the concentration of RBD and S1 antibody reactivity in the positive donor control, and used titrations of the donor pool included on each ABCORA plate to calculate the S1 and RBD content of plasma samples in relation to it. We used the same strategy in combination with the WHO International Standard Anti-SARS-CoV-2 Immunoglobulin (NIBSC 20/136[32]) to defer IU/ml content of the internal ABCORA positive donor pool and the individual specimen tested (Supplementary Fig. 7). The WHO International Standard consists of a pool of plasma from individuals with confirmed SARS-CoV-2 infection. RBD and S1 content of the ABCORA positive donor pool quantified via the polyclonal WHO standard was highly similar within each Ig class (Supplementary Fig. 7b). In contrast, RBD values estimated by the mAb CR3022 were a factor 2.4–3.9 lower than the corresponding S1 values, suggesting an affinity difference of CR3022 for the two antigens (Supplementary Fig. 7b).

**Temporal evolution of SARS-CoV-2 binding antibody response**. Antibody binding of 140 convalescent patients was measured longitudinally in 274 measurements with ABCORA 2.0, including 251 measurements from 120 patients with known time since positive RT-PCR. We assumed the antibodies (analyzed as logMFI-FOE) were declining with time from 21 days after positive RT-PCR and estimated the decay using a power law mixed model with random effect on the intercept[62]. As time measures days post first positive RT-PCR result (Fig. 5b) or days post onset of symptoms (Supplementary Fig. 11b) were employed. Half-lives (t1/2, in days) of significant response with negative decays were calculated based on the respective estimated decay parameters. Analyses were performed in R version 3.6.3 using packages lme4 and lmerTest[63].

**SARS-CoV-2 pseudo-neutralization assay**. SARS-CoV-2 plasma neutralization activity was defined using an HIV-based pseudovirus system as described[33]. The env-inactivated HIV-1 reporter construct pHIV-1NL4-3 ΔEnv-NanoLuc (pHIV-1Nanoluc) and HT1080/ACE2cl.14 cells were kindly provided by P. Bieniasz, Rockefeller University, NY, USA. To create a SARS-CoV-2 spike expression plasmid (P_CoV2_Wuhan), a codon-optimized C terminal truncated (AA 1255-1273) spike encoding gene of strain Wuhan-Hu-1 (GenBank accession no. MN908947) was synthesized (GeneArt, Thermo Fisher Scientific, Waltham, MA) and cloned into pcDNA3.1. Pseudotyped SARS-CoV-2 spike expressing viruses were generated by co-transfecting 293-T cells with a mixture of pHIV-1Nanoluc, P_CoV2_Wuhan and PEI Max (Polysciences Europe GmbH, Hirschberg, Germany). After 48 h virus supernatants were filtered (0.2 μm) and stored in aliquots at −80 °C until use. Infectivity of virus stocks was measured by infection of HT1080/ACE2cl.14 cells. For this 384-well culture plate pretreated with poly-L-Lysine were seeded with HT1080/ACE2cl.14 (2200 cells/well) one day before the assay. Cells were infected with titrated virus stocks and NanoLuc luciferase activity in cell lysates measured 48 h post infection using the Nano-Glo Luciferase Assay System (Promega, Fitchburg, WI). For this, cells were washed once with PBS, supernatant was removed and cells were lysed with 20 μl/well of Luciferase Cell Lysis reagent (Promega, Fitchburg, WI) for 15 min under continuous shaking at room temperature. 20 μl of 1/50 diluted NanoGlo buffer were added and NanoLuc luciferase activity (relative light units, RLU) was measured after 5 min incubation at room temperature on a Perkin Elmer EnVision reader. Input of SARS-CoV-2 pseudoviruses for neutralization assays was adjusted to yield virus infectivity corresponding to 5–10 × 10^6 RLU (corresponding to 100-250-fold over background RLU values) in the absence of inhibitors. To measure plasma neutralization activity six serial 4-fold dilutions of plasma starting at a 1/25 dilution were prepared. 20 μl of the diluted plasma and 20 μl of virus were preincubated for 1 h at 37 °C and then 30 μl of the virus/plasma mix were transferred to 384-well plates seeded with HT1080/ACE2cl.14 cells in a volume of 30 μl. This resulted in a final concentration of the plasma starting dilution of 1/100. Plasma neutralization titers causing 50%, 80% and 90% reduction in viral infectivity (NT50, NT80 and NT90, respectively) compared to controls without plasma were calculated by fitting a sigmoid dose–response curve (variable slope) to the RLU data, using GraphPad Prism with constraints (bottom = 0, top = 100). If 50% inhibition was not achieved at the lowest plasma dilution of 1/100, a 'less than' value was recorded. All measurements were conducted in duplicates.

**Predicting neutralization based on ABCORA binding activity**. To compare the ability of SARS-CoV-2 binding activity measured in ABCORA 2.0 to predict the neutralization status, we measured neutralization activity to Wuhan-Hu-1 in SARS-CoV-2 positive individuals ($N = 467$) and classified individuals as high neutralizers (NT50 > 250, $N = 332$) and low neutralizers (NT50 < 250, $N = 135$). Six different classification models were designed to assign individuals to the high or low neutralizers category, based on their ABCORA2.0 binding patterns we established two univariable logistic regression (ULR) models that included mean MFI-FOE spike antigen S1 reactivities and mean MFI-FOE spike antigen RBD reactivities, respectively. S1 and RBD were chosen due to their highest correlation with NT50 ($r = 0.82$ and $r = 0.80$ for the total spike reactivities of S1 and RBD respectively, Supplementary Fig. 9). In addition to these two ULR, we established a multivariable logistic regression model including both mean S1 and RBD reactivities (MFI). We further devised three models that considered all 12 SARS-CoV-2 binding parameters recorded by ABCORA 2.0. Two multivariable logistic regression models were based on a principal component analysis on all binding activities and included the first two (respectively four) principal components, which explained 60% (respectively 75%) of the variance in the data. We also included in our model comparison a classification based on a random forest analysis that incorporated all 12 SARS-CoV-2 binding activity variables.

For all six models, performance was assessed in 100 cross-validation sets: each set was built by randomly sampling without replacement among the 467 measurements available. 80% of the data set was used to train the model ($N = 374$). Prediction of neutralization status was realized on the other 20% ($N = 93$) and compared to the true NT50 value and neutralization status, using a roc curve. The area under the curve (AUC) was computed for all six models in each cross-validation run. The Bayesian information criterion (BIC) was computed for all five logistic regressions in each cross-validation run.

To increase the utility of the ULR-S1 prediction model for clinical diagnostics we devised a modified neutralization prediction model ULR-S1-SOC based on the

signal over cut-off (SOC) values reported for ABCORA 2.0. The ULR-S1-SOC estimates the probability of NT50 > 250 based on the sum of S1 SOC values for IgG, IgA and IgM as indicated in Eq. (2).

$$P(NT50>250) = \frac{\exp(a + b * log10(sum\,S1\,SOC))}{1 + \exp(a + b * log10(sum\,S1\,SOC))} \qquad (2)$$

With estimated values: $a = -2.6447$ and $b = 3.5353$.

ULR-S1-SOC estimates for the probability of NT50 > 100 were analyzed in analogy (Supplementary Table 9 and Supplementary Fig. 10).

**Association between HCoVs and SARS-CoV-2 reactivities**. To explore the association between HCoVs and SARS-CoV-2 reactivities, we defined a new HCoV response variable (HCoV high/low) for each antibody class (IgG, IgA, IgM) as follows: a patient had high HCoV Ig reactivity for a given antibody class if its measurements were higher than the population median in at least three out of the four HCoV measurements (HKU1, OC43, NL63, 229E). To assess inter-dependencies between HCoV and SARS-CoV-2 responses, we then included the HCoV response variable in a linear regression model of SARS-CoV-2 reactivities in the same antibody class. The linear regression models were estimated on a subset of SARS-CoV-2 positive patients ($N = 204$), measured on ABCORA 5.0 less than 60 days since positive RT-PCR. The restriction to 60 days was chosen to allow modeling the effect of time with splines. This time period restriction further guaranteed a gender balance, as convalescent donors with longer follow up were all males recruited for a plasma therapy study (CPT-ZHP, Swissmedic 2020TpP1004). Regression analyses were adjusted on time (days post positive RT-PCR or onset of symptoms; as a spline with 3 degrees of freedom), age (as a spline with 3 degrees of freedom) and gender.

Among the 204 patients analyzed for interdependencies, information regarding hospitalization status (not hospitalized, hospitalized not in ICU, hospitalized in ICU) was known for 160 of them. For 80 patients we had samples that were collected less than 30 days since SARS-CoV-2 diagnosis by positive RT-PCR allowing an estimate of HCoV levels close to SARS-CoV-2 acquisition. We performed an additional analysis on this subset of patients using an ordinal regression and a logistic regression to predict the hospitalization status depending on high or low IgG HCoV reactivity, adjusted on age (as a spline with 3 degrees of freedom) and gender. We checked for robustness of the result in a sensitivity analysis by adjusting on time since positive RT-PCR in addition to age and gender (Supplementary Fig. 15).

**Statistical analysis**. Statistical analyses were performed in R (Version 3.6.3). Figures were made using the ggplot2 package[64]. When included, boxplots represent the following: median with the middle line, upper and lower quartiles with the box limits, 1.5x interquartile ranges with the whiskers and outliers with points. Heat-maps were made using the ComplexHeatmap package[65]. Significance of Spearman rank correlations were assessed through asymptotic $t$ approximation. Differences in means between two groups with independent measures were tested using two-sided $t$-tests. When applicable, multiple testing was adjusted using Bonferroni correction for multiple comparisons. A one-way ANOVA with 3 degrees of freedom was used in addition to two-tailed $t$-tests in Fig. 6 to provide insights on overall versus group comparison. When analyzing datasets including repeated measurements of the same individuals (Fig. 3a, Fig. 5b, Fig. 5c, Supplementary Fig. 8a, Supplementary Fig. 11b), we used linear and power-law mixed models with time since positive RT-PCR or time since symptom onset (continuous or binary variable) as fixed effect and individual as random effect. In the case of Fig. 5c and Supplementary Fig. 11c, the decreasing slope of neutralization titers was estimated by considering only individuals with neutralization titers above the detection levels (NT50 > 100) at their first measurement. For all linear mixed models, a Satterthwaite approximation for a two-sided $t$-test was used to determine if the estimated slope was significantly different from 0. In addition, half-lives were obtained from the decay rate estimated on the log of either MFI-FOEs (Fig. 5b, Supplementary Fig. 11b) or NT50s (Fig. 5c, Supplementary Fig. 11c) as follows: $t1/2 = t0 * \exp(\log(10)2)/rate)$, with $t0 = 21$ days since positive RT-PCR or since symptom onset. In Fig. 7b, c, linear regressions were used to estimate the association between HCoV and SARS-CoV-2 reactivities: a Student $t$-test with two-sided hypothesis was used to assess if this association was significantly different from 0.

**Reporting summary**. Further information on research design is available in the Nature Research Reporting Summary linked to this article.

## Data availability
The raw serological measurements generated in this study are provided in Supplementary Data 2. Data depicted in charts and graphs are made available in the Source Data file. Source data are provided with this paper.

## Code availability
Codes to assess serostatus based on the ABCORA 2.3 method are available at: https://github.com/chlpasin/SARS-CoV-2-serology[66].

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

## Acknowledgements

This work was supported by a grant of the Pandemiefonds of the University of Zurich Foundation to A.T., a grant of the Swiss Red Cross to B.F. and A.T., a grant of the University Hospital Zurich Innovation Grant to M.G.M., the Swiss National Science Foundation grant 31CA30_196906 to H.F.G., A.T., R.K. and the Gilead COVID-19 RFP Research Program COMMIT Grant #: IN-SW-983-6078 (to H.F.G., A.T., R.K.). I.A.A. is supported by a research grant of the Promedica Foundation. Roche Diagnostics supported the study with providing test material for a proportion of the Elecsys S tests. We thank the staff of the Institute of Medical Virology diagnostics unit, sample triage and administration for their support and the staff of the participating clinics for coordinating the sample collection in the frame of the included clinical trials.

## Author contributions

I.A.A., C.P., M.S., A.T., H.F.G. and R.D.K. conceived and designed the study and analyzed data. I.A.A., M.S., M.M.S., S.E., M.C.H., L.M., M.E.S., A.H., A.A., C.R.N., J.B., M.H. designed and performed binding antibody experiments. P.R., J.W. and S.E. conducted neutralization experiments and analyzed data. S.S. developed an analysis app. C.P. and R.D.K. analyzed data. D.L.B., M.M., A.W., S.K.R., B.M.F., E.S., J.G., C.B., P.M.M.S., M.G.M., H.F.G., A.W., U.K., J.B. and M.H. were involved in patient recruitment, provided samples from study and diagnostic repositories and analyzed patient data. A.T., I.A.A., C.P., and M.S. wrote the paper, which all coauthors commented on.

## Competing interests

The authors declare no competing interests.
