## [Peer Review File · Nature Communications]

REVIEWER COMMENTS

Reviewer #1 (COVID-19 serology/Ab responses, computational) (Remarks to the Author):

Abela et al study binding antibody responses by Luminex to CoV-2 and hCovs in large cohorts of CoV-2 infected people and controls. They argue that they can model the responses to predict neutralisation and that higher HCoV Ab responses during CoV-2 infection alters the Ab pattern of CoV-2 responses. Overall I felt the novelty and innovation was modest compared to the claims being made.

Comments

The authors state line 220 "we concluded that the MFI-FOE readout at 1/100 provides an excellent estimate for the S1 and RBD antibody content in plasma and that this straightforward and easily obtainable readout". The authors must live in an interesting world if a Flexmap machine and conjugated fluorescent beads represents an "easily obtainable readout". I note later in the discussion neut assays are described as "have limited turnover and scalability" and that this assay with multiple antigens and detectors requiring a machine costing >\$130,000 is said to "obliterate(e) the need to conduct labor-intensive and costly neutralization tests in clinical diagnostics". This came across as a pitch to poorly informed investors.

The HCoV infections in the immunocompromised group – presumably the immunocompromised state restricted their ability to mount Ab responses? Its unclear how this helps as a control for cross-reactivity. Only HKU1 was in the antigen panel – how did this specifically fare in the 17 subjects with HKU1 infection.

The authors describe binding and neut data as "closely linked" although the best r value of many many conducted is ~0.7. This is a rather modest surrogate and these claims should be tempered. The authors use a binary neut titre of 250 to first study their computational serology. Although it varies with different assays, 250 is generally a pretty high titre in convalescent subjects, with only about a ¼ of their cohort in this strata. A "protective" level of neut ab is probably in the range of 10-30 according to recent analyses (D. Khoury et al Medrxiv 10.1101/2021.03.09.21252641v1) . Thus although the model defined high neut subjects to some degree, the system is probably unhelpful at a clinical level (even if it were practical).

The longitudinal data shows a biphasic decay in several responses. This has previously been modelled (Ref 2 in the manuscript). An overall half life underestimates the early decay and overestimates the late decay.

The MFI values are very low to the hCovs - usually 1-10 at a level where there is a lot of noise in these assays. Many of the apparently significant differences look at best a few fold – I would not call this "varies considerably".

Fig 7a, if I understand it correctly, shows lower NL63 and 229E S1 IgG responses were observed in CoV-2 infected subjects compared to uninfected subjects. Later in this results section it is said "interdependencies between HCoV IgG and SARS-CoV-2 specific IgG were only observed for the S2 response." Were these S1 or S2? Previous data shows cross reactivity with hCovs S, especially S2, after CoV-2 infection.

The interpretation that prior hCoV S2 immunity and CoV-2 responses in Fig 7b was confusing. It doesn't seem surprising that hCoV responses are associated with CoV-2 responses or vice versa following infection since there is some cross reactivity. Without serology prior to infection I don't see how this data supports the claim that prior HCov immunity is modulating CoV-2 responses.

Minor comment: What is the x-axis in Fig 2A? The correlation lines should be removed in Fig2b and elsewhere.

Reviewer #2 (biostats)

(no comment for the authors; only comment for the editor; suggesting randomizing the dataset for validation)

Reviewer #3 (computational) Remarks to the Author):

I read the article, but I am unsure whether I am a good fit for the review. The part with the logistic regression and the random forests are ok from a methodological point of view as they used a training and independent test set. However, I cannot judge the assays used in the study, which are, however, crucial here.

Reviewer #4 (COVID-19 serology) (Remarks to the Author):

In the study "Multifactorial SARS-CoV-2 seroprofiling dissects interdependencies with human coronaviruses and predicts neutralization activity", IA Abela and colleagues, describe a new serological test based on the parallel assessment of anti-SARS-CoV-2 and anti-human coronavirus (HCoV) circulating antibody. The assay named ABCORA was tested for its capacity to evaluate seropositivity, neutralization and antibody response dynamics.

The study, which involves the analysis of large sample sets is definitely interesting, however the following point should be addressed:

1. Results and methods are described extensively. The authors could deliver better the main message and conclusion of their study condensing the text. I would suggest to revise the paper avoiding repetition of information and figures (eg. Are Fig 3. and Supplementary figure 8 the same? Figure 5 and and supplementary Figure 10 also share a panel).
2. Abstract: " Intriguingly, HCoV reactivity was significantly higher in SARS-CoV-2 negative donors. Amongst SARS-CoV-2 infected individuals, elevated SARS-CoV- 2 responses were linked to higher HCoV activity suggesting that pre-existing HCoV immunity may confer protection against SARS-CoV-2 acquisition and promote development of SARS-CoV-2 specific antibody responses".  Looking at the available clinical information regarding the individuals in the study, it would be interesting to know if those having high HCoV immunity and then elevated SARS-CoV-2 response, had milder symptoms than others, to support further the hypothesis.

Samples' demographics other than age and gender would be useful for data interpretation (Supplementary Table 11).

3. Clarify line 121: "In pre-pandemic individuals, IgM binding to SARS-CoV-2 and HKU1 was highly correlated whereas IgG and IgA reactivates were less affected (Supplementary Fig. 5b). . Line 131 again..."Indeed, pre-pandemic children showed a higher correlation of IgM HKU1 and SARS 135 CoV-2 (Supplementary Fig. 5c)... These data underline that cross-reactivity needs to be considered but mostly affects low level responses as evidenced by the measurements in pre-pandemic individuals (Fig. 1a, b, Supplementary Fig. 5). It would be important to clarify if this is due simply to a correlation between low signals (artefact) or if it a real correlation, because this could affect the interpretation of all the following data. Which is the raw level of MFI?

NCOMMS-21-14181 Point-to-point reply

Reviewer #1

Query 1: *The authors state line 220 “we concluded that the MFI-FOE readout at 1/100 provides an excellent estimate for the S1 and RBD antibody content in plasma and that this straightforward and easily obtainable readout”. The authors must live in an interesting world if a Flexmap machine and conjugated fluorescent beads represents an “easily obtainable readout”. I note later in the discussion neut assays are described as “have limited turnover and scalability” and that this assay with multiple antigens and detectors requiring a machine costing >\$130,000 is said to “obliterate(e) the need to conduct labor-intensive and costly neutralization tests in clinical diagnostics”. This came across as a pitch to poorly informed investors.*

Reply: This is a misunderstanding to some extent. With “easily attainable” we refer here to the fact that the assay does not require a plasma titration to be measured as the single 1/100 dilutions gives a very good measure of the antibody content. A single sample for which the measurement can be used straight versus a 6- 8 serial dilution for which an EC50 has to be defined, is - we are certain the reviewer will agree - an easier and more straightforward option. We rephrased the sentence to make this clear.

In respect to the utility of the bead-based assay over neutralization assays the reviewer questions, this is definitely a matter of different worlds how one views this. While straightforward to implement in research, cell-based neutralization assays are unfortunately not a valid option for diagnostics,. Daily service with this type of assay is simply impossible to maintain. Neutralization assays are also not cheap if the entire work-flow and man-power is considered. Readers required for neutralization assays are not commonly used in diagnostics and equally costly. The multifactorial assessment the ABCORA tests allows including the prediction of neutralization responses, has proven extremely useful in our own diagnostics laboratory and is intensively used by clinicians dealing with complex SARS-CoV-2 cases for more than a year now. Anyway, that is our “real world” experience which, we fully agree, is not needed to be included in the write up of our study, as the focus is on the biological findings. We therefore adjusted the respective passages.

Query 2: *The HCoV infections in the immunocompromised group – presumably the immunocompromised state restricted their ability to mount Ab responses? Its unclear how this helps as a control for cross-reactivity. Only HKU1 was in the antigen panel – how did this specifically fare in the 17 subjects with HKU1 infection.*

Reply: Immunocompromised individuals have historically been the most frequently tested for HCoV infection when reporting respiratory infections and are also now among the most heavily monitored for SARS-CoV-2. The inclusion of these individuals as a control group is therefore important from a clinical/diagnostics perspective. We explained the diagnostic importance of this control group in the methods section (line 425ff original ms/ line 422ff revised ms). As we point out, because of their immunosuppression, immune responses to HCoV might be expected to be lower in these individuals. However, it may also be that the responses are present but less mature. Nevertheless, since some

individuals experience recurrent infections, it cannot be ruled out that higher HCoV reactivity may also occur in some cases. Lastly, seroreactivity to vaccination is increasingly being monitored in immunocompromised individuals to guide the need for booster vaccination. Inclusion of immunocompromised individuals and exclusion of cross-reactivity in this group is therefore critical. As our results show, we found no evidence of higher SARS-CoV-2 reactivity in immunocompromised individuals, allowing the ABCORA test to be used in all samples without prior knowledge of patient history. Measurements of ABCORA 2 which include HKU1 only are depicted in Figure 1. ABCORA 5 measurements including all four HCoV are shown in Supplementary Figure 12. As suggested by reviewer 1 we now also show in Supplementary Figure 4 HCoV reactivity patterns in immune compromised individuals alongside information of the infecting HCoV.

Query 3: *The authors describe binding and neut data as “closely linked” although the best r value of many many conducted is ~0.7. This is a rather modest surrogate and these claims should be tempered.*

Reply: We adjusted the passage and related figure titles.

Query 4: *The authors use a binary neut titre of 250 to first study their computational serology. Although it varies with different assays, 250 is generally a pretty high titre in convalescent subjects, with only about a ¼ of their cohort in this strata. A “protective” level of neut ab is probably in the range of 10-30 according to recent analyses (D. Khoury et al Medrxiv 10.1101/2021.03.09.21252641v1). Thus although the model defined high neut subjects to some degree, the system is probably unhelpful at a clinical level (even if it were practical).*

Reply: We thank the reviewer for noting an inconsistency in the text. Strata counts of high and low neutralizers were unfortunately flipped in the text but the correct distribution was displayed in the corresponding figure. 75% of our cohort were of course in the strata, 25% not. We corrected the respective passage in the text.

Please note: Titers in our neutralization assay system correspond to the final assay dilution, whereas Khoury et al. refer to the serum pre-dilution. Hence, titers of 10-30 reported by Khoury et al correspond to 40-120 in our assay system. It is important to note that overall the convalescent individuals studied in Khoury et al, have a comparatively lower level of neutralization activity as we see. In our cohort the median titer was 671 (corresponding to 168 if calculated as in Khoury et al). Restricting to patients with early convalescence (<50 days post infection as in Khoury et al), our cohort portrayed a median titer of 348 (corresponding to 87 as calculated by Khoury et al). Khoury et al observed in their cohort a comparable median titer of 52. We opted for a higher cut-off in assessing neutralization capacity as low neutralization titers against the vaccine strain do commonly not provide protection against SARS-CoV-2 VOC, particularly the Delta variant as shown by several studies. We therefore considered tailoring our analysis to a lower end cut-off as less informative as it will no longer bear clinical and epidemiological implication against current circulating variants. However, to ease comparison with other studies as suggested by reviewer 1, we now include additional data using a titer of 1:100 (equaling 1: 25 in the Khoury study) as strata. This new data is depicted in supplementary figure 10 and shows exactly the same pattern as we had seen before with the higher threshold. Together these analyses demonstrate the validity of our findings and highlight that the observed relationship with neutralization that does not depend on a specific threshold level.

Query 5: *The longitudinal data shows a biphasic decay in several responses. This has previously been modelled (Ref 2 in the manuscript). An overall half-life underestimates the early decay and overestimates the late decay.*

Reply: We thank the reviewer for this comment. We explored alternative ways of modeling the antibody decay by comparing the simple exponential decay we initially used, a biphasic model with two slopes used by Ref 2, and a power-law model which is also used to model antibody decline after infection (Fraser et al. Vaccine. 2007 May 22;25(21):4324-33). By comparing the models using the Akaike Information Criterion (AIC) and assuming decline from 21 days post infection, and exploring several potential times of slope change in the biphasic model, we found that the power law provided the best fit for the majority of the 12 SARS-CoV-2 parameters probed. While the biphasic model yielded slightly better AIC values IgA RBD and S1, the model proposed a biologically implausible sharp decrease followed by an increase in titers for these two antibody responses. We therefore chose the power law model as the most appropriate in the setting of our study and adapted the respective figures (Figures 5b, 5c and Supplementary Figures 11b, 11c) and half-life estimation accordingly.

Query 6: *The MFI values are very low to the hCovs - usually 1-10 at a level where there is a lot of noise in these assays. Many of the apparently significant differences look at best a few fold – I would not call this “varies considerably”.*

Reply: We were surprised to see such a variation to a recurring infection, but yes, overall the magnitude is low. We adjusted the wording to make this clearer.

Query 7: *Fig 7a, if I understand it correctly, shows lower NL63 and 229E S1 IgG responses were observed in CoV-2 infected subjects compared to uninfected subjects. Later in this results section it is said “interdependencies between HCoV IgG and SARS-CoV-2 specific IgG were only observed for the S2 response.” Were these S1 or S2? Previous data shows cross reactivity with hCovs S, especially S2, after CoV-2 infection.*

Reply: We thank the reviewer for noting that this passage is difficult to follow. Yes, we saw that for IgG responses in infected individuals only S2 IgG and HCoV IgG were associated. S1 IgG and RBD IgG showed no significant association. For IgA and IgM reactivity with all antigens (S1, RBD, S2 and N) were linked with HCoV activity. The fact that S2 IgG were also significantly associated with high hCoV activity is particular intriguing, considering the recent report by Loyal et al. (Science first release; DOI: 10.1126/science.abh1823) that highlight cross-reactive HCoV T helper responses and their impact on modulating SARS-CoV-2 antibody responses including S2 antibodies. We reworded the respective passages in the results and discussion section to make their content clearer and added a reference to the Loyal et al. study to highlight the implications of our findings.

Query 8: *The interpretation that prior hCoV S2 immunity and CoV-2 responses in Fig 7b was confusing. It doesn't seem surprising that hCoV responses are associated with CoV-2 responses or vice versa following infection since there is some cross reactivity. Without serology prior to infection I don't see how this data supports the claim that prior HCov immunity is modulating CoV-2 responses.*

Reply: We appreciate that these interdependencies are not easy to grasp on first glance. We first show in Figure 6 that in samples collected in the same time period, uninfected individuals have higher HCoV levels than infected. Thus, HCoV reactivity appears to have a protective effect against infection. The next question we addressed was whether HCoV levels have an impact on the development of SARS-CoV-2 responses. We employed here a cross-sectional analysis that controlled for time of infection. Sera derived shortly prior to infection to retrieve baseline HCoV levels as the reviewer requests, were not available. As we show, there are good other means to study the impact of pre-existing HCoV n SARS-CoV-2 responses. By simply stratifying the HCoV response into high and low responders, we were able to show that individuals with high HCoV reactivity also developed higher SARS-CoV-2 antibodies to multiple antigens. Thus, although HCoV reactivity in infected is lower than in uninfected, once infected, individuals with higher HCoV reactivity mount a higher SARS-CoV-2 response. The fact that we see this pattern across antigens supports the validity of our finding. Importantly, the effect is strongest for IgM and IgA responses, indicating a boost of early responses. In agreement with findings recently reported by Loyal et al (Science, DOI: 10.1126/science.abh1823), we find a clear difference for IgG S2 responses between individuals with high and low HCoV set points. This strongly suggest that SARS-CoV-2 S2 IgG responses, supported by the S2 T helper response defined by Loyal et al, have a particular advantage in maturing rapidly. To assist future readers of our work we reworded the relevant sections to explain our findings and their implications better. In addition, prompted by a request of reviewer 4, we now include new data that show a striking impact of pre-existing HCoV immunity on disease severity. This analysis shows that individuals with low HCoV reactivity are at higher risk for developing severe disease requiring intensive care. Collectively, these data suggest a certain protective role of HCoV immunity that needs to be considered.

Query 9: *Minor comment: What is the x-axis in Fig 2A? The correlation lines should be removed in Fig2b and elsewhere.*

Reply: We adjusted the axis as requested. After discussion with colleagues we opted however to keep the correlation lines, as these were genuinely found as useful for orientation and interpretation and there are no methodological reasons not to show them.

Reviewer #2

Query 10: *suggesting randomizing the dataset for validation*

Reply: As suggested by reviewer 2 we include and now show an additional control analysis with randomization (see new supplementary Table 6). The outcome of this analysis fully confirmed and validated our prior findings.

Reviewer #3

I read the article, but I am unsure whether I am a good fit for the review. The part with the logistic

regression and the random forests are ok from a methodological point of view as they used a training and independent test set. However, I cannot judge the assays used in the study, which are, however, crucial here.

Reply: We thank reviewer 3 for the positive assessment of our methodological approach.

Reviewer #4

In the study “Multifactorial SARS-CoV-2 seroprofiling dissects interdependencies with human coronaviruses and predicts neutralization activity”, IA Abela and colleagues, describe a new serological test based on the parallel assessment of anti-SARS-CoV-2 and anti-human coronavirus (HCoV) circulating antibody. The assay named ABCORA was tested for its capacity to evaluate seropositivity, neutralization and antibody response dynamics.

The study, which involves the analysis of large sample sets is definitely interesting, however the following point should be addressed:

Reply: We thank the reviewer for the positive assessment of our work.

Query 11: *Results and methods are described extensively. The authors could deliver better the main message and conclusion of their study condensing the text. I would suggest to revise the paper avoiding repetition of information and figures (eg. Are Fig 3. and Supplementary figure 8 the same? Figure 5 and supplementary Figure 10 also share a panel).*

Reply: We thank the reviewer for the suggestions of condensing the text to shift more focus on the main findings. We are confident that the revised manuscript has improved with this. As to the figures the reviewer noted as repetitive: These are not identical, they show data presented as time post PCR diagnosis versus time since symptom onset. From a clinical perspective considering both of these different reference time points is of considerable interest and we would therefore argue to keep both analyses in the paper.

Query 12: *2. Abstract: “ Intriguingly, HCoV reactivity was significantly higher in SARS-CoV-2 negative donors. Amongst SARS-CoV-2 infected individuals, elevated SARS-CoV- 2 responses were linked to higher HCoV activity suggesting that pre-existing HCoV immunity may confer protection against SARS-CoV-2 acquisition and promote development of SARS-CoV-2 specific antibody responses”.  Looking at the available clinical information regarding the individuals in the study, it would be interesting to know if those having high HCoV immunity and then elevated SARS-CoV-2 response, had milder symptoms than others, to support further the hypothesis.*

Reply: We agree that a detailed analysis based on symptoms severity would be very interesting. Samples included in our study were from different clinical trials and subject to distinct ethics approval. Therefore, individual symptoms description was not available. We had however information for a subgroup of individuals whether they were hospitalized or, not and whether hospitalized individuals required intensive care at the sampling time point. As hospitalization is a

clear indicator for disease severity, we used this information to address the reviewer's question whether pre-existing HCoV may have an impact on shaping disease severity. Indeed, as we show now in the new Figure 8, we observed both a higher risk of hospitalization, and, among hospitalized patients, a higher risk for requiring intensive care among individuals with low HCoV responses. Thus these results further support a potential protective effect of pre-existing HCoV immunity.

Query 13: *Samples' demographics other than age and gender would be useful for data interpretation (Supplementary Table 11).*

Reply: Samples included in our study were from different clinical trials and subject to distinct ethics approval due to which only selected information on demographics were available. We extended Supplementary Table 11 (now Supplementary Table 12) as requested to include the information on all variables we had access to for the current study, these were, age, gender, time post PCR diagnosis, time post onset of symptoms, hospitalization status.

Query 14: *Clarify line 121: "In pre-pandemic individuals, IgM binding to SARS-CoV-2 and HKU1 was highly correlated whereas IgG and IgA reactivates were less affected (Supplementary Fig. 5b). Line 131 again..." Indeed, pre-pandemic children showed a higher correlation of IgM HKU1 and SARS 135 CoV-2 (Supplementary Fig. 5c)... These data underline that cross-reactivity needs to be considered but mostly affects low level responses as evidenced by the measurements in pre-pandemic individuals (Fig. 1a, b, Supplementary Fig. 5). It would be important to clarify if this is due simply to a correlation between low signals (artefact) or if it a real correlation, because this could affect the interpretation of all the following data. Which is the raw level of MFI?*

Reply: In these analyses we investigated the level of cross-reactivity of HCoV with SARS-CoV-2 antigens in pre-pandemic individuals. As we hoped, the cross-reactivity was low. Raw MFI-FOE levels of all measurements are depicted in Figures 1a and Supplementary Figure 12. The low level of cross-reactivity observed allowed the development of the highly specific ABCORA test. Yet, as we point out cross-reactivity exists and needs to be considered for the data interpretation in different ways. First, when exposure to SARS-CoV-2 is assessed based on serological data, cross-reactive HCoV antibodies need to be accounted for. To allow calling at highest possibly specificity we introduced two different analysis strategies for the ABCORA test, one based on threshold setting, the other on a random forest computation, which both allow an accurate differentiation between SARS-CoV-2 specific and cross-reactive responses. Later in the manuscript, we focus on the cross-reactivity not as an unwanted effect in the assay but for its biological relevance. These analyses highlight, that HCoV cross-reactivity has implications and is not an artefact.

REVIEWERS' COMMENTS

Reviewer #1 (Remarks to the Author):

I am satisfied with the response.

Reviewer #4 (Remarks to the Author):

The authors answered carefully to each query raised by the reviewers clarifying some aspects and improved the paper, I don't have further comments.